# Iterative Substructure Extraction for Molecular Relational Learning with Interactive Graph Information Bottleneck

**Shuai Zhang**[1,2*] **Junfeng Fang**[1*] **Xuqiang Li**[1,3] **Hongxin Xiang**[5] **Jun Xia**[6]
**Ye Wei**[7] **Wenjie Du**[1,3,4†] **Yang Wang**[1,3,4†]
[1] University of Science and Technology of China (USTC)
[2] School of Computer Science and Technology, USTC
[3] Suzhou Institute for Advanced Research, USTC
[4] State Key Laboratory of Precision and Intelligent Chemistry, USTC
[5] Hunan University    [6] Westlake University    [7] City university of Hong Kong
{shuaizhang, xuqiangli, duwenjie}@mail.ustc.edu.cn
{fangjf1997, yeweiastronomer}@gmail.com
xianghx@hnu.edu.cn   xiajun@westlake.edu.cn   angyan@ustc.edu.cn

## Abstract

Molecular relational learning (MRL) seeks to understand the interaction behaviors between molecules, a pivotal task in domains such as drug discovery and materials science. Recently, extracting core substructures and modeling their interactions have emerged as mainstream approaches within machine learning-assisted methods. However, these methods still exhibit some limitations, such as insufficient consideration of molecular interactions or capturing substructures that include excessive noise, which hampers precise core substructure extraction. To address these challenges, we present an integrated dynamic framework called Iterative Substructure Extraction (ISE). ISE employs the Expectation-Maximization (EM) algorithm for MRL tasks, where the core substructures of interacting molecules are treated as latent variables and model parameters, respectively. Through iterative refinement, ISE gradually narrows the interactions from the entire molecular structures to just the core substructures. Moreover, to ensure the extracted substructures are concise and compact, we propose the Interactive Graph Information Bottleneck (IGIB) theory, which focuses on capturing the most influential yet minimal interactive substructures. In summary, our approach, guided by the IGIB theory, achieves precise substructure extraction within the ISE framework and is encapsulated in the **IGIB-ISE**. Extensive experiments validate the superiority of our model over state-of-the-art baselines across various tasks in terms of accuracy, generalizability, and interpretability. Our code can be found at https://github.com/congcijueqi/IGIB-ISE.

## 1 Introduction

Molecular relational learning (MRL) Rozemberczki et al. (2021) Fang et al. (2024) Du et al. (2024) aims to represent interaction properties between molecules, such as potential drug-drug interactions (DDI) Xiong et al. (2022), chromophores Ye et al. (2021) in different solvents *etc.*, which has gained significant attention. The core substructure of molecules embodies the essence of their physicochemical properties in molecular interactions Chi et al. (2010); Bender & Glen (2004). As shown in Figure 1 (a), styrene oxide exhibits primarily blue fluorescence in hexane due to its epoxide moiety, while in acetonitrile, the fluorescence shifts to a yellowish hue due to the influence of its vinyl group. For capturing interaction behavior between molecules, current models often rely on the chemical prior that core substructures encapsulate key characteristics of molecular, *i.e. the linchpin* Book (2014); Jerry (1992).

---

*Equal Contribution
†Corresponding Author

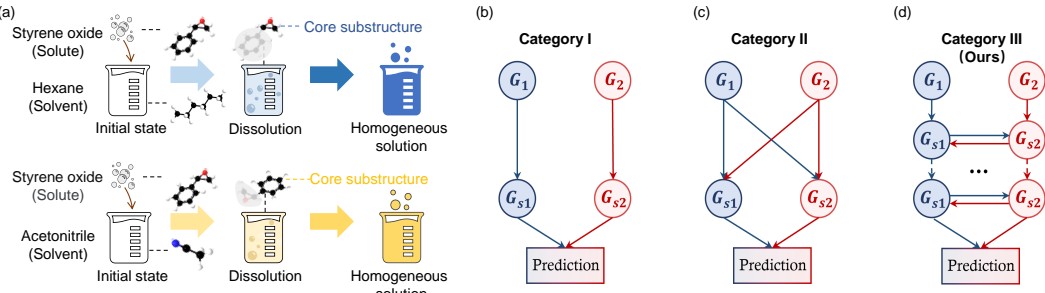

Figure 1: (a) shows the differences in fluorescence emission of styrene oxide in various solvents, while (b), (c), and (d) compare different paradigms for MRL. Best viewed in color.

In order to accurately mine these vital substructures, prevailing methodologies can be broadly categorized into two categories. **Category I** (Figure 1 (b)) exemplified by models such as SSI-DDI Nyamabo et al. (2021)and STNN-DDI Yu et al. (2022a), which individually obtain substructures from each molecule before subsequent interaction. However, such methods present notable limitations. Primarily, the isolated extraction of substructures from each molecular neglects the potential interplay between different molecular substructures. This overlooks the fact that the selection of substructures for one molecule can be significantly influenced by another, depending on the specific task Lang et al. (2021). This leads to a somewhat superficial understanding of MRL, failing to fully grasp the dynamic and interconnected nature of molecular interactions in the biochemical context Silverman & Holladay (2014); Böhm et al. (2004); Schneider et al. (2018).

In sight of this, **Category II** (Figure 1 (c)) Li et al. (2023) have pivoted towards a more holistic approach to address these limitations. These methods simultaneously consider a second molecule as a conditional factor during the generation of a molecular substructure Lee et al. (2023a). This paradigm shift ensures that the substructure generation is not an isolated event but an interactive process. However, such methods also present their challenges. Considering that core substructures often play a crucial role in molecular interactions Jia et al. (2009); Nyamabo et al. (2021), integrating the complete profile of an interacting molecule into the substructure generation can be overwhelming. It carries the risk of compromising generalizability and the inclusion of redundant information Lee et al. (2023b); Tang et al. (2023), particularly for molecules that share similar structures yet exhibit significant functional divergence in specific combinations, *e.g.*, Activity Cliffs Tamura et al. (2023); Van Tilborg et al. (2022); Schneider et al. (2018).

Considering these factors, we aim to harness the interaction effects of core substructures to facilitate the process of interactive substructure extraction. In this paper, we propose the **I**terative **S**ubstructure **E**xtraction (**ISE**) framework. As shown in Figure 1 (d), ISE employs the Expectation-Maximization Dempster et al. (1977) (EM) algorithm to iteratively uncover core interactive substructures between molecular pairs, where two molecular core substructures are regarded as latent variables and model parameters, respectively. Under the premise of molecular interactions and inherent symmetry, ISE facilitates iterative interaction and substructure selection between the two graphs, ultimately identifying the optimal core substructure combination. This ensures that the extracted substructures depend solely on the core substructures of another molecule, thereby minimizing the influence of extraneous structures and enhancing alignment with the essence of molecular interactions.

Furthermore, to ensure that the ISE framework obtains concise and compact interactive substructures, we draw inspiration from the Graph Information Bottleneck (GIB) theory Wu et al. (2020a), a method used to extract core substructure-based compressed variable information from a single input graph. We introduce the **I**nteractive **G**raph **I**nformation **B**ottleneck (**IGIB**) to ensure comprehensive consideration of substructure information from another graph during the process of substructure compression, achieved through the introduction of conditional mutual information. IGIB lays down a theoretical foundation and establishes a precise optimization goal for the analysis of biochemical molecule interactions and the mining of interactive substructures.

Our contributions can be summarized as follows:

- We identify and articulate the limitations of existing Molecular Relational Learning methods in addressing the problem of core substructure extraction. For the first time, we redefine this problem in the context of the Expectation-Maximization (EM) algorithm and propose the ISE framework, which employs iterative coupling of substructures to optimize the extraction process.

- We introduce IGIB as a theoretical foundation for Molecular Relational Learning, more consistent with chemical principles. IGIB-ISE is highly compatible with our ISE framework, representing a paradigm shift that emphasizes the importance of inter-substructure dynamics in capturing the essence of molecular interactions more effectively.

- The superiority of our approach is empirically validated through extensive experiments on multiple Molecular Relational Learning datasets. Our method outperforms existing approaches in terms of accuracy, generalizability, and interpretability in MRL. Notably, our ISE framework revitalizes the interpretable research of core substructures, shedding light on the selection process of essential interactive substructures.

## 2 PRELIMINARIES

In this section, we first formally describe the problem formulation (Section 2.1). Then, we introduce the Graph Information Bottleneck (GIB) theory Wu et al. (2020b) (Section 2.2). Finally, we introduce the key variables and the execution process in the EM algorithm (Section 2.3).

### 2.1 PROBLEM PREDEFINITION

A molecule can be naturally represented as a graph, $\mathcal{G} = (\mathcal{V}, \mathcal{E})$, where $\mathcal{V}$ represents the set of nodes and $\mathcal{E}$ denotes the set of bonds. In the context of MRL, for each data point in the dataset, we receive a pair of molecular graphs, $\mathcal{G}_1$ and $\mathcal{G}_2$ as input, along with their associated label $\mathbf{Y}$. The label $\mathbf{Y}$ is a scalar value, i.e., $\mathbf{Y} \in (-\infty, \infty)$, for molecular interaction prediction tasks, while it is a binary class label, i.e., $\mathbf{Y} \in \{0, 1\}$, for the binary classification task. $\mathcal{G}_{s1}$ and $\mathcal{G}_{s2}$ represent the subgraph of $\mathcal{G}_1$ and $\mathcal{G}_2$, respectively, with subgraphs corresponding to the molecular substructures.

### 2.2 GRAPH INFORMATION BOTTLENECK

In graph-related tasks, discerning which substructures within a graph are significant and which are not is essential. The GIB method addresses this challenge by learning a bottleneck graph $\mathcal{G}_{\text{IB}}$ for a given graph $\mathcal{G}$. This approach compresses the source graph to retain the structures pertinent to predicting the target random variable while discarding those irrelevant to the target Yu et al. (2020; 2022b); Miao et al. (2022).

**Definition 2.2 (GIB):** Given an input graph $\mathcal{G}$ and label $\mathbf{Y}$, GIB aims to extract a compact subgraph $\mathcal{G}_{IB}$, while keeping the information relevant for predicting $\mathbf{Y}$ by optimizing the following objective:

$$\arg\min_{\mathcal{G}_{\text{IB}}} -I(\mathbf{Y}; \mathcal{G}_{\text{IB}}) + \beta I(\mathcal{G}; \mathcal{G}_{\text{IB}}) \tag{1}$$

where $I(\cdot, \cdot)$ denotes the mutual information Tishby et al. (2000) between random variables. $\beta$ is a Lagrangian multiplier for balancing the two mutual information terms.

### 2.3 EXPECTATION-MAXIMIZATION ALGORITHM

The ISE framework utilizes the EM algorithm Dempster et al. (1977). The EM algorithm is designed to handle situations involving **observed variables**, **latent variables**, **model parameters**, and their distributions. It iteratively estimates parameters in probabilistic models, with a particular focus on latent variables. This iterative process consists of two steps: the **E-step**, which computes the posterior probabilities of latent variables, and the **M-step**, where model parameters are updated by maximizing the likelihood using expected values obtained from the E-step.

## 3 METHODOLOGY

In this section, we introduce our proposed method. First, we define the Interactive Graph Information Bottleneck (IGIB) (Section 3.1). Then, we detail the application of the EM algorithm within the ISE

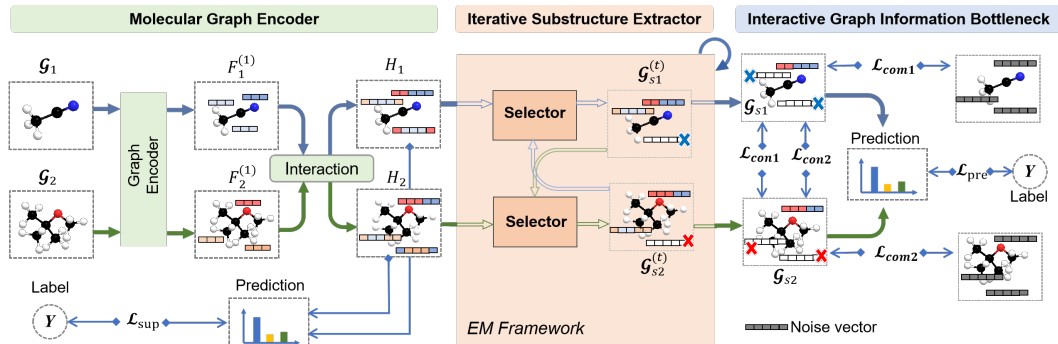

Figure 2: The overall framework of our proposed IGIB-ISE. Best viewed in color.

(Section 3.2). Next, we present the architecture of interactive substructure extraction (Section 3.3). Finally, we describe the entire model optimization process based on IGIB (Section 3.4).

## 3.1 THE THEORY OF INTERACTIVE GRAPH INFORMATION BOTTLENECK

We introduce the IGIB theory to guide interactive substructure extraction in MRL tasks. Specifically, for two input graphs $\mathcal{G}_1$ and $\mathcal{G}_2$ and their interaction label $\mathbf{Y}$, IGIB posits that the generation process of the interactive substructures $\mathcal{G}_{s1}$ and $\mathcal{G}_{s2}$ should not only maximize the mutual information between the substructures and the target output $\mathbf{Y}$, but also minimize the mutual information between $\mathcal{G}_{s1}$ and the original graph $\mathcal{G}_1$ when conditioned on $\mathcal{G}_2$ (and vice versa for $\mathcal{G}_{s2}$). Notably, the condition here is based on the substructure rather than the entire original graph as in CGIB Lee et al. (2023a), thereby mitigating the influence of redundant and irrelevant information. We formalize this theory as Definition 3.1 and refer to it as IGIB.

**Definition 3.3 (IGIB):** Given a pair of graphs $\mathcal{G}_1$ and $\mathcal{G}_2$ and their label information $\mathbf{Y}$, IGIB aims to extract a pair of compact yet maximally informative substructures $\mathcal{G}_{s1}$ and $\mathcal{G}_{s2}$, which are related to each other by optimizing the following objective:

$$\arg\min_{\mathcal{G}_{s1}, \mathcal{G}_{s2}} -I\left(\mathbf{Y}; \mathcal{G}_{s1}, \mathcal{G}_{s2}\right) + \beta_1 I\left(\mathcal{G}_1; \mathcal{G}_{s1} \mid \mathcal{G}_2\right) + \beta_2 I\left(\mathcal{G}_2; \mathcal{G}_{s2} \mid \mathcal{G}_{s1}\right), \tag{2}$$

where $\beta_1$ and $\beta_2$ are trade-off parameters. Note that the two parameters $\beta_1$ and $\beta_2$ incorporated by the above equation are designed to adapt to the unique scenarios in MRL where the interaction between two molecules is not entirely symmetrical.

By focusing on the essential information within the substructures and minimizing extraneous features, IGIB provides a more efficient and task-specific way of handling the complexities inherent in molecular interaction modeling.

## 3.2 EM ALGORITHM FOR ITERATIVE SUBSTRUCTURE EXTRACTION

**The assignment of variables:** From the perspective of the EM algorithm, we re-examine the relationship between input molecular graphs $\mathcal{G}_1$, $\mathcal{G}_2$, substructures $\mathcal{G}_{s1}$, $\mathcal{G}_{s2}$, and label $\mathbf{Y}$. Firstly, $\mathcal{G}_1$, $\mathcal{G}_2$, and $\mathbf{Y}$ can be directly provided by the dataset. Therefore, $\mathbf{Y}_{\mathcal{G}}$ is regarded as the **observed variables**, including $\mathcal{G}_1$, $\mathcal{G}_2$, and $\mathbf{Y}$. Secondly, because most interactions between molecules arise from the interactions between core substructures, the contribution of $\mathcal{G}_1$ and $\mathcal{G}_2$ to $\mathbf{Y}$ is through $\mathcal{G}_{s1}$ and $\mathcal{G}_{s2}$ (substructures). We designate $\mathcal{G}_{s2}$ as the **latent variables**. For $\mathcal{G}_{s1}$, considering that the core interacting substructures between molecules influence each other, the latent variables $\mathcal{G}_{s2}$ are influenced by the substructure $\mathcal{G}_{s1}$. Thus, $\mathcal{G}_{s1}$ is regarded as the **model parameters** (vice versa).

**Iterative process: E-step:** Estimate the latent variables $\mathcal{G}_{s2}$ while freezing the model parameters $\mathcal{G}_{s1}$, then utilize it to calculate the Evidence Lower Bound (ELBO). **M-step:** Refine the optimal model parameters $\mathcal{G}_{s1}$ which maximizes the ELBO obtained in the E-step. The procedure marked by green lines in the EM framework of Figure 2 represents the E-step, while the process indicated by blue lines corresponds to the M-step. Their optimization targets are defined as follows, where $(t)$ is denoted by the iteration step.

- **Initialization:** Given two molecular graphs $\mathcal{G}_1$ and $\mathcal{G}_2$, set $\mathcal{G}_{s1}^{(0)} = \mathcal{G}_1$.

- **E-step:** Estimate the substructure $\mathcal{G}_{s2}^{(t)}$ according to $\mathcal{G}_{s1}^{(t)}$, and then calculate the ELBO:

$$\text{ELBO} \rightarrow E_{\mathcal{G}_{s2}^{(t)} | \mathbf{Y}_\mathcal{G}, \mathcal{G}_{s1}^{(t)}} [\log \frac{P(\mathcal{G}_{s2}^{(t)}, \mathbf{Y}_\mathcal{G} \mid \mathcal{G}_{s1}^{(t)})}{P(\mathcal{G}_{s2}^{(t)} \mid \mathbf{Y}_\mathcal{G}, \mathcal{G}_{s1}^{(t)})}]; \tag{3}$$

- **M-step:** Find the corresponding substructure $\mathcal{G}_{s1}^{(t+1)}$ that maximizes the above ELBO:

$$\mathcal{G}_{s1}^{(t+1)} := \arg\max_{\mathcal{G}_{s1}} E_{\mathcal{G}_{s2}^{(t)} | \mathbf{Y}_\mathcal{G}, \mathcal{G}_{s1}^{(t)}} [\log \frac{P(\mathcal{G}_{s2}^{(t)}, \mathbf{Y}_\mathcal{G} \mid \mathcal{G}_{s1}^{(t)})}{P(\mathcal{G}_{s2}^{(t)} \mid \mathbf{Y}_\mathcal{G}, \mathcal{G}_{s1}^{(t)})}]; \tag{4}$$

- **Output:** Iteratively execute E-step and M-step, then output the interactive substructures $\mathcal{G}_{s1}$ and $\mathcal{G}_{s2}$.

For the detailed derivation of E-step and M-step, please refer to Appendix B.2 and B.3. Convergence proof of ISE framework is provided in Appendix B.4.

## 3.3 ARCHITECTURE OF INTERACTIVE SUBSTRUCTURE EXTRACTION

The architecture of interactive substructure extraction consists of the molecular graph encoder and iterative substructure extractor.

**Molecular Graph Encoder.** For the molecular graphs $\mathcal{G}_1 = (\mathcal{V}_1, \mathcal{E}_1)$ and $\mathcal{G}_2 = (\mathcal{V}_2, \mathcal{E}_2)$, we employ GNN for their encoding:

$$F_1^{(1)} = \text{GNN}(\mathcal{V}_1, \mathcal{E}_1), \qquad F_2^{(1)} = \text{GNN}(\mathcal{V}_2, \mathcal{E}_2), \tag{5}$$

where $F_1^{(1)} \in \mathbb{R}^{N^1 \times d}$ and $F_2^{(1)} \in \mathbb{R}^{N^1 \times d}$ are the node embedding matrices for $\mathcal{G}_1$ and $\mathcal{G}_2$, respectively. Next, we focus on expanding node features. This expansion is based on the interaction architecture of CIGIN Pathak et al. (2020). To facilitate the interaction between two graphs, the graph-graph interaction map $\mathbf{I} \in \mathbb{R}^{N^1 \times N^2}$ is computed by using the following equations: $\mathbf{I}_{ij} = \text{sim}(F_{1i}^{(1)}, F_{2j}^{(1)})$, where $\text{sim}(\cdot, \cdot)$ denotes the cosine similarity. Here, $N^1$ and $N^2$ represent the number of nodes in $\mathcal{G}_1$ and $\mathcal{G}_2$, respectively. Next, we compute the embedding matrices $F_1^{(2)} \in \mathbb{R}^{N^1 \times d}$ and $F_2^{(2)} \in \mathbb{R}^{N^2 \times d}$, each embedding matrix incorporating information from its paired graph. These matrices are derived based on the interaction map as follows: $F_1^{(2)} = \mathbf{I} \cdot F_2^{(1)}, \quad F_2^{(2)} = \mathbf{I}^\top \cdot F_1^{(1)}$, where $(\cdot)$ denotes matrix multiplication. Based on these, the aggregation operation of node features can be completed as follows: $H_1 = F_1^{(1)} || F_1^{(2)}, H_2 = F_2^{(1)} || F_2^{(2)}$, where $H_1$ and $H_2$ are the final node embedding features of $\mathcal{G}_1$ and $\mathcal{G}_2$, and $||$ denotes the concatenation operation.

**Iterative Substructure Extractor.** Initially, we initialize $\mathcal{G}_{s1}^{(0)} = \mathcal{G}_1$ and $H_{s1}^{(0)} = H_1$, where $H_{s1}$ represents the node embedding features of $\mathcal{G}_{s1}$.

**E-step:** The interaction operation employed in molecular graph encoder along with a two-layer Multi-Layer Perceptron (MLP) is utilized to assess the importance of each node in $\mathcal{G}_2$, as expressed by:

$$\mathbf{I}_{ij}^{(t)} = \text{sim}(H_{s1i}^{(t-1)}, H_{2j}), \quad P^{(t)} = \text{Sigmoid}\left(\text{MLP}\left((\mathbf{I}^{(t)^\top} \cdot H_1)\right)\right), \tag{6}$$

Inspired by the theorys of information bottlenecks Tishby et al. (2000) and focusing on node significance, we introduce random noise into nodes to facilitate substructure extraction, as suggested by Yu et al. (2022b). This process involves the following operations:

$$h_i^{(t)} = \lambda_i^{(t)} h_i + (1 - \lambda_i^{(t)}) \epsilon, \tag{7}$$

$$\lambda_i^{(t)} = \text{Sigmoid}\left(\frac{1}{\tau} \log\left(\frac{p_i^{(t)}}{1 - p_i^{(t)}}\right) + \log\left(\frac{u}{1 - u}\right)\right), \tag{8}$$

where $i$ represents the node number in $\mathcal{G}_2$, $h_i$ represents the embedding feature of node $i$, $\lambda_i^{(t)}$ is drawn from a Bernoulli distribution with probability $p_i^{(t)}$. To ensure differentiability in the sampling

process, we adopt the Gumbel sigmoid Maddison et al. (2016); Jang et al. (2016) for the discrete random variable $\lambda_i^{(t)}$. The transmission probability $p_i^{(t)}$ regulates information flow from $h_i$ to $h_i^{(t)}$. The parameter $\tau$ adjusts sensitivity to noise randomness, and $u$ is drawn from a uniform distribution, $u \sim \text{Uniform}(0, 1)$. Thus, the interactive substructure $\mathcal{G}_{s2}^{(t)}$ from $\mathcal{G}_2$ is successfully extracted because nodes excluded in this process have been injected with noise, diluting their inherent information.

**M-step:** The interactive substructure $\mathcal{G}_{s1}^{(t+1)}$ is obtained based on $\mathcal{G}_{s2}^{(t)}$ to achieve maximum likelihood. Due to the symmetric nature of molecular interactions in MRL, we employ the same network architecture as in the E-step to determine $\mathcal{G}_{s1}^{(t+1)}$. Finally, upon completion of the iterations, the set2set network Vinyals et al. (2015) is utilized to pool the substructures $\mathcal{G}_{s1}$ and $\mathcal{G}_{s2}$, resulting in the substructure representation vectors $z_{\mathcal{G}_{s1}}$ and $z_{\mathcal{G}_{s2}}$. These vectors serve as compact representations that encode the essential information of the substructures for further prediction.

### 3.4 Model Optimization Based on IGIB

We provide the upper bound of the intended Definition 3.1, which should be minimized during training.

Minimizing $-I\left(\mathbf{Y}; \mathcal{G}_{s1}, \mathcal{G}_{s2}\right)$: We consider $P_\theta\left(\mathbf{Y} \mid \mathcal{G}_{s1}, \mathcal{G}_{s2}\right)$ as the variational estimation of $P\left(\mathbf{Y} \mid \mathcal{G}_{s1}, \mathcal{G}_{s2}\right)$. Thus, we derive:

$$
\begin{aligned}
I\left(\mathbf{Y}; \mathcal{G}_{s1}, \mathcal{G}_{s2}\right) &\geq \mathbb{E}_{(\mathbf{Y}, \mathcal{G}_{s1}, \mathcal{G}_{s2})} \log\left[\frac{P_\theta\left(\mathbf{Y} \mid \mathcal{G}_{s1}, \mathcal{G}_{s2}\right)}{P(\mathbf{Y})}\right] \\
&= \mathbb{E}_{(\mathbf{Y}, \mathcal{G}_{s1}, \mathcal{G}_{s2})} \log\left[P_\theta\left(\mathbf{Y} \mid \mathcal{G}_{s1}, \mathcal{G}_{s2}\right)\right] + H(\mathbf{Y}) := \mathcal{L}_{pre},
\end{aligned}
\tag{9}
$$

where $H(\mathbf{Y})$ is constant across all data, it will be omitted in the model optimization process.

Minimizing $I\left(\mathcal{G}_1; \mathcal{G}_{s1} \mid \mathcal{G}_{s2}\right)$: Based on the chain rule of mutual information, we decompose it into:

$$
I\left(\mathcal{G}_1; \mathcal{G}_{s1} \mid \mathcal{G}_{s2}\right) = I\left(\mathcal{G}_{s1}; \mathcal{G}_1, \mathcal{G}_{s2}\right) - I\left(\mathcal{G}_{s1}; \mathcal{G}_{s2}\right).
\tag{10}
$$

For $I(\mathcal{G}_{s1}; \mathcal{G}_1, \mathcal{G}_{s2})$, $z_{\mathcal{G}_{s1}}$ represents the encoding of $\mathcal{G}_{s1}$. Minimizing $I(\mathcal{G}_{s1}; \mathcal{G}_1, \mathcal{G}_{s2})$ is equivalent to minimizing $I(z_{\mathcal{G}_{s1}}; \mathcal{G}_1, \mathcal{G}_{s2})$. We approximate $I\left(z_{\mathcal{G}_{s1}}; \mathcal{G}_1, \mathcal{G}_{s2}\right)$ using a variational inference approach $q(z_{\mathcal{G}_{s1}})$ as an estimate for $p(z_{\mathcal{G}_{s1}})$.

$$
\begin{aligned}
I\left(z_{\mathcal{G}_{s1}}; \mathcal{G}_1, \mathcal{G}_{s2}\right) &= \mathbb{E}_{\left(z_{\mathcal{G}_{s1}}, \mathcal{G}_1, \mathcal{G}_{s2}\right)} \log\left[\frac{p_\Phi\left(z_{\mathcal{G}_{s1}} \mid \mathcal{G}_1, \mathcal{G}_{s2}\right)}{p\left(z_{\mathcal{G}_{s1}}\right)}\right] \\
&= \mathbb{E}_{(\mathcal{G}_1, \mathcal{G}_{s2})} \log\left[\frac{p_\Phi\left(z_{\mathcal{G}_{s1}} \mid \mathcal{G}_1, \mathcal{G}_{s2}\right)}{q\left(z_{\mathcal{G}_{s1}}\right)}\right] - \mathbb{E}_{\left(z_{\mathcal{G}_{s1}}, \mathcal{G}_1, \mathcal{G}_{s2}\right)} KL\left(p\left(z_{\mathcal{G}_{s1}}\right) \| q\left(z_{\mathcal{G}_{s1}}\right)\right).
\end{aligned}
\tag{11}
$$

Here, the function $p_\Phi$ refers to the objective of the process described in Section 3.3, which is based on the EM algorithm and aims to generate $\mathcal{G}_{s1}$. Given the non-negativity of the Kullback-Leibler divergence, it follows that:

$$
I\left(z_{\mathcal{G}_{s1}}; \mathcal{G}_1, \mathcal{G}_{s2}\right) \leq \mathbb{E}_{(\mathcal{G}_1, \mathcal{G}_{s2})} KL\left(p_\Phi\left(z_{\mathcal{G}_{s1}} \mid \mathcal{G}_1, \mathcal{G}_{s2}\right) \| q\left(z_{\mathcal{G}_{s1}}\right)\right) := \mathcal{L}_{com1}.
\tag{12}
$$

For the term $I\left(\mathcal{G}_{s1}; \mathcal{G}_{s2}\right)$, it is necessary to augment the mutual information between $\mathcal{G}_{s1}$ and $\mathcal{G}_{s2}$. To achieve this, we employ a contrastive loss function Tian et al. (2020); Hjelm et al. (2018), which has been demonstrated to effectively increase mutual information. The contrastive loss is defined as:

$$
\mathcal{L}_{con1} = -\frac{1}{K} \sum_{i=1}^{K} \log \frac{\exp(\text{sim}(z_{\mathcal{G}_{s1}}^i, z_{\mathcal{G}_{s2}}^i)/\tau)}{\sum_{j=1, j\neq i}^{K} \exp(\text{sim}(z_{\mathcal{G}_{s1}}^i, z_{\mathcal{G}_{s2}}^j)/\tau)},
\tag{13}
$$

where $\text{sim}(\cdot, \cdot)$ denotes a similarity function, the superscript denotes different pairs of molecules, and $\tau$ is a temperature parameter employed to adjust sensitivity to the similarity between samples.

Minimizing $I\left(\mathcal{G}_2; \mathcal{G}_{s2} \mid \mathcal{G}_{s1}\right)$: The upper bound of the objective function is obtained similarly to minimizing $I\left(\mathcal{G}_1; \mathcal{G}_{s1} \mid \mathcal{G}_{s2}\right)$, leading to the derivation of additional loss functions $\mathcal{L}_{com2}$ and $\mathcal{L}_{con2}$:

$$
\mathcal{L}_{com2} := \mathbb{E}_{(\mathcal{G}_2, \mathcal{G}_{s1})} KL\left(p_\Phi\left(z_{\mathcal{G}_{s2}} \mid \mathcal{G}_2, \mathcal{G}_{s1}\right) \| q\left(z_{\mathcal{G}_{s2}}\right)\right),
\tag{14}
$$

$$\mathcal{L}_{con2} = -\frac{1}{K} \sum_{i=1}^{K} \log \frac{\exp(\text{sim}(z_{\mathcal{G}_{s2}}^i, z_{\mathcal{G}_{s1}}^i)/\tau)}{\sum_{j=1, j\neq i}^{K} \exp(\text{sim}(z_{\mathcal{G}_{s2}}^i, z_{\mathcal{G}_{s1}}^j)/\tau)}. \tag{15}$$

In summary, our overall loss function equation 16, serving as an upper bound for equation 2, is constructed by combining these components to optimize our IGIB-ISE:

$$\mathcal{L}_{sum} = \mathcal{L}_{pre} + \beta_1(\mathcal{L}_{com1} + \mathcal{L}_{con1}) + \beta_2(\mathcal{L}_{com2} + \mathcal{L}_{con2}) + \mathcal{L}_{sup}, \tag{16}$$

where $\beta_1$ and $\beta_2$ is the trade-off parameters. $\mathcal{L}_{sup}$ is the prediction loss between label $\mathbf{Y}$ and the pair of input graphs $(\mathcal{G}_1, \mathcal{G}_2)$. Here, $\mathcal{L}_{pre}$ can be modeled as the cross entropy loss for classification and the mean square loss for regression. $\mathcal{L}_{com1}$ and $\mathcal{L}_{com2}$ represent KL divergence between extracted interactive substructures and the noise graph, encouraging substructure compression. $\mathcal{L}_{con1}$ and $\mathcal{L}_{con2}$ denote contrastive loss between two substructures to reinforce their relationship. The detailed proofs for $\mathcal{L}_{pre}$ and $\mathcal{L}_{com1}$ will be provided in Appendix B.5 and B.6.

## 4 EXPERIMENT

In this section, we conduct extensive experiments to answer the following questions:

- **RQ1:** Can our model enhance the performance of molecular relational learning tasks?
- **RQ2:** Does the interactive extraction of substructures improve the performance of IGIB-ISE?
- **RQ3:** How effective is the ISE module in terms of interpretability?

### 4.1 EXPERIMENTAL SETTINGS

In this section, we briefly introduce the datasets, baselines, and evaluation metrics. More details on experimental settings, dataset descriptions, baseline introductions, hyper-parameter selection, and the performance of the models on additional evaluation metrics are provided in Appendix D.3 and 7.

**Datasets.** To evaluate the performance of our model, we conduct experiments based on nine datasets. For the molecular interaction prediction task, we utilize the **Chromophore** dataset Joung et al. (2020), including absorption, emission, and excited state lifetime. Additionally, **MNSol** Marenich et al. (2020), **FreeSolv** Mobley & Guthrie (2014), **CompSol** Moine et al. (2017), **Abraham** Grubbs et al. (2010), and **CombiSolv** Vermeire & Green (2021a) are also considered, which describe the solvation free energy for a solute-solvent pair. For the drug-drug interaction prediction task, we incorporate three DDI datasets, including **ZhangDDI** Zhang et al. (2017), **ChChMiner** Zitnik et al. (2018) and **DeepDDI** Zitnik et al. (2018), which record the adverse reactions between drug-drug pairs.

**Baselines.** For both tasks, we compare our method with the diverse SOTA models that could be regarded as three categories, as shown in Figure 1. CGIB Lee et al. (2023a), CMRL Lee et al. (2023b), and CIGIN Pathak et al. (2020) have widely proven their superiority on MRL. For the drug-drug interaction prediction tasks, we chose routine SSI-DDI Nyamabo et al. (2021), GoGNN Wang et al. (2020), DSN-DDI Li et al. (2023) and MHCADDI Deac et al. (2019). For the molecular interaction prediction tasks, we chose additional models D-MPNN Vermeire & Green (2021a), Explainable GNN Low et al. (2022), and UNI-MOL Zhou et al. (2023), due to the single-task nature of these baseline models.

**Evaluation metrics.** The performance of the molecular interaction prediction task is evaluated by RMSE Pathak et al. (2020), while the DDI prediction task is evaluated in terms of classification accuracy Wang et al. (2021).

### 4.2 PREDICTION PERFORMANCE (RQ1)

The empirical performance of our model is summarized in Table 1 and Table 2, respectively. Our observations are as follows:

**Obs.1: IGIB-ISE outperforms other baselines in both molecular interaction prediction and drug-drug interaction prediction tasks.** The experimental results in Table 1 and Table 2 (a) illustrate

Table 1: Performance on molecular interaction prediction task (regression) in terms of RMSE.

| Model | Chromophore | | | MNSol | FreeSolv | CompSol | Abraham | CombiSolv |
|---|---|---|---|---|---|---|---|---|
| | Absorption | Emission | Lifetime | | | | | |
| **Category I** | | | | | | | | |
| Explainable GNN | 22.74 (1.06) | 28.09 (1.43) | 0.834 (0.017) | 0.673 (0.024) | 1.258 (0.044) | 0.353 (0.012) | 0.751 (0.034) | 0.417 (0.049) |
| UNI-MOL | − − | − − | − − | 0.657 (0.019) | 1.210 (0.041) | 0.339 (0.014) | 0.672 (0.028) | 0.629 (0.011) |
| CIGIN | 19.47 (0.34) | 25.17 (0.29) | 0.815 (0.011) | 0.644 (0.022) | 1.013 (0.013) | 0.301 (0.016) | 0.435 (0.010) | 0.498 (0.009) |
| D-MPNN | 24.08 (1.47) | 29.34 (0.93) | 0.829 (0.022) | 0.667 (0.017) | 1.107 (0.031) | 0.353 (0.013) | 0.608 (0.033) | 0.559 (0.006) |
| **Category II** | | | | | | | | |
| CGIB | 18.05 (0.34) | 24.62 (0.29) | 0.783 (0.009) | 0.613 (0.023) | 0.918 (0.029) | 0.279 (0.016) | 0.386 (0.010) | 0.440 (0.009) |
| CMRL | 18.24 (0.31) | 25.69 (0.27) | 0.791 (0.008) | 0.623 (0.019) | 0.927 (0.024) | 0.274 (0.014) | 0.375 (0.008) | 0.423 (0.006) |
| **Category III** | | | | | | | | |
| ISE | 17.81 (0.37) | 24.66 (0.42) | 0.773 (0.026) | 0.607 (0.028) | 0.825 (0.039) | 0.268 (0.013) | 0.369 (0.014) | 0.400 (0.010) |
| **IGIB-ISE** | **16.90** (0.32) | **23.83** (0.26) | **0.747** (0.015) | **0.572** (0.024) | **0.713** (0.034) | **0.266** (0.010) | **0.343** (0.009) | **0.394** (0.008) |

Table 2: Performance on drug-drug interaction prediction task (classification) in terms of ACC.

| Model | (a) Transductive | | | (b) Inductive Setting 1 | | | (c) Inductive Setting 2 | | |
|---|---|---|---|---|---|---|---|---|---|
| | ZhangDDI | ChChMiner | DeepDDI | ZhangDDI | ChChMiner | DeepDDI | ZhangDDI | ChChMiner | DeepDDI |
| **Category I** | | | | | | | | | |
| GoGNN | 84.14 (0.46) | 91.17 (0.46) | 93.54 (0.35) | 61.51 (1.87) | 67.48 (1.56) | 67.53 (1.52) | 57.37 (3.27) | 64.27 (4.31) | 63.96 (3.64) |
| CIGIN | 85.98 (0.30) | 92.71 (0.32) | 93.29 (0.47) | 65.27 (1.24) | 76.35 (0.92) | 71.84 (0.89) | 57.11 (1.75) | 64.25 (2.33) | 65.54 (2.93) |
| SSI-DDI | 86.97 (0.27) | 93.26 (0.24) | 94.27 (0.25) | 62.38 (1.53) | 76.94 (1.32) | 69.77 (0.86) | 57.24 (2.38) | 65.61 (2.51) | 66.53 (3.53) |
| MHCADDI | 77.86 (0.59) | 84.26 (0.64) | 87.01 (0.77) | 61.81 (1.27) | 65.77 (0.76) | 63.94 (0.98) | 57.84 (2.28) | 59.24 (5.39) | 61.17 (3.67) |
| **Category II** | | | | | | | | | |
| CMRL | 87.78 (0.37) | 94.43 (0.25) | 95.99 (0.34) | 68.38 (1.12) | 80.54 (0.66) | 74.12 (0.55) | 59.53 (1.37) | 67.09 (1.54) | 68.29 (1.78) |
| CGIB | 87.69 (0.73) | 94.68 (0.35) | 95.76 (0.72) | 68.34 (0.66) | 80.67 (0.77) | 74.29 (0.53) | 58.39 (2.04) | 68.78 (1.84) | 68.26 (1.39) |
| DSN-DDI | 87.65 (0.13) | 94.23 (0.26) | 93.37 (0.34) | 67.68 (0.87) | 79.94 (0.72) | 74.35 (0.62) | 59.11 (1.42) | 68.36 (1.54) | 69.17 (1.28) |
| **Category III** | | | | | | | | | |
| ISE | 88.45 (0.42) | 94.82 (0.67) | 96.13 (0.057) | 67.63 (1.02) | 80.42 (0.98) | 74.56 (0.83) | 58.87 (1.49) | 69.27 (1.93) | 69.42 (1.54) |
| **IGIB-ISE** | **88.84** (0.32) | **95.56** (0.28) | **96.65** (0.37) | **68.75** (0.83) | **81.15** (0.79) | **75.28** (0.69) | **59.96** (1.23) | **70.34** (1.53) | **70.54** (1.27) |

significant improvements. We argue that our model architecture, which iteratively extracts interactive substructures, can extract more accurate substructures. This ensures that more precise information can be provided in subsequent prediction tasks, thereby enhancing the performance of the model.

**Obs.2: Our model also demonstrates excellent generalization performance in various inductive settings.** We conducted additional experiments in two inductive settings: Setting 1 ensures that at least one drug is unseen in the test dataset while Setting 2 ensures that both test molecules are unlearned (as shown in Table 2 (b) and (c)). IGIB-ISE achieves higher prediction accuracy across three datasets, showcasing its superior generalization capabilities. This heightened performance can be attributed to the extensive interaction among captured core substructures, underscoring IGIB-ISE's practical utility, especially in handling emerging drug molecules.

**Obs.3: The performance exhibits a clear ascending order for Category I, Category II, and our model.** Compared to models lacking intermolecular interactions (Category I), models considering molecular interactions exhibit clear superiority. Furthermore, the performance of IGIB-ISE surpasses all evaluation metrics for models of Category II across all datasets. This suggests that considering substructure interactions between molecules significantly enhances model performance. The introduction of the EM algorithm enables the substructure to evolve dynamically through interactive iterations. The substructure can be accurately represented during the iterative process, and a detailed analysis of its dynamic iteration process is provided in Appendix E.6.

**Obs.4: The IGIB theory effectively improves model performance.** Specifically, results from the IGIB-ISE framework demonstrate superior performance compared to using ISE alone. Guided by the IGIB theory, the ISE framework's ability to extract generalized and accurate interactive substructures is significantly improved. This enhancement is attributed to IGIB's proactive nature, which promotes the compression of substructures to their fullest extent. This compression process yields more concise structures, substantially enhancing the model's capabilities. Detailed results and analyses are provided in Appendix E.2.

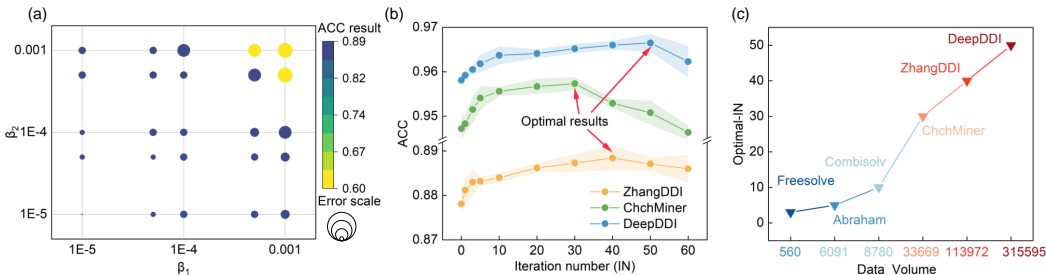

Figure 3: Hyperparameter experimental results. (a) Test results under different values of $\beta_1$ and $\beta_2$. (b) Test results under different iteration numbers (IN); (c) the optimal-IN on various datasets.

### 4.3 ABLATION AND SENSITIVITY EXPERIMENT (RQ2)

In this section, we analyze the core hyperparameters, including $\beta_1$, $\beta_2$, and iteration numbers (IN) of the EM algorithm, as illustrated in Figure 3.

**Obs.5: There exists the optimal point of $\beta_1 = 0.0001$ and $\beta_2 = 0.0001$ in terms of the model performance.** $\beta_1$ and $\beta_2$ govern the delicate balance between prediction accuracy and information compression within the two molecular graphs. As shown in Figure 3 (a), setting $\beta_1 = 0.001$ and $\beta_2 = 0.001$ results in suboptimal model performance. This can be attributed to the aggressive compression encouraged by these values, resulting in the model inadequately capturing the molecular information crucial for the target task. Conversely, the reduction of $\beta_1$ and $\beta_2$ implies preservation of more original graph information. However, it does not consistently guarantee improved performance. This is because, in such scenarios, the model may struggle to identify the precise interactive substructure necessary for accurate predictions, thereby compromising its generalization capabilities.

**Obs.6: The performance of the model is closely related to the IN.** In Figure 3 (b), we observe that as IN increases, the model's performance gradually improves on all datasets. This improvement can be attributed to our iterative extraction method, which enhances the accuracy of the extracted substructures. However, as IN becomes larger, the performance on the test set begins to decline. This is because excessive iterations lead to the extraction of 'locally optimal' substructures, thereby diminishing generalization performance.

**Obs.7: The optimal IN value for the model increases with the size of the dataset.** As illustrated in Figure 3(c), the optimal IN value of the model increases with the size of the dataset. We attribute this phenomenon to the distributional error during the dataset partitioning process. When the dataset is smaller, the distributional error is larger, whereas it would decrease as the dataset size increases. Therefore, by dynamically adjusting the iteration times of the ISE framework, we can enable the model to achieve different levels of fitting and generalization capabilities, demonstrating the robust adaptability of our model to diverse datasets.

### 4.4 INTERPRETABILITY ANALYSIS (RQ3)

In this section, we visually analyze the DDI prediction process based on IGIB-ISE. We conducted a random selection of four drug pairs, all of which could generate a DDI reaction. The dynamic nature of substructure identification renders the ISE module distinctly remarkable.

**Obs.8: IGIB-ISE exhibits distinct core substructure recognition in molecule pairs interaction.** Illustrated in Figure 4 (a) and 4 (b), each graph demonstrates the substructure selection results for Theophylline and Hydroxyurea molecules when interacting with different molecules to produce DDI. Taking the example of the molecular pairs formed by Theophylline with Domperidone and Atazanavir, significant differences in the selection of O and N are observed. This reflects that the model can extract core substructures for different molecular combinations.

**Obs.9: Dynamic iterative substructure extraction enhances core substructure learning.** For different numbers of iterations, as the iterative updates continue along with the iterative selection of substructures, the learning process of ISE towards core substructures is enhanced. As depicted in Figure 4 (c), the acyl group is gradually phased out, while other groups progressively manifest their

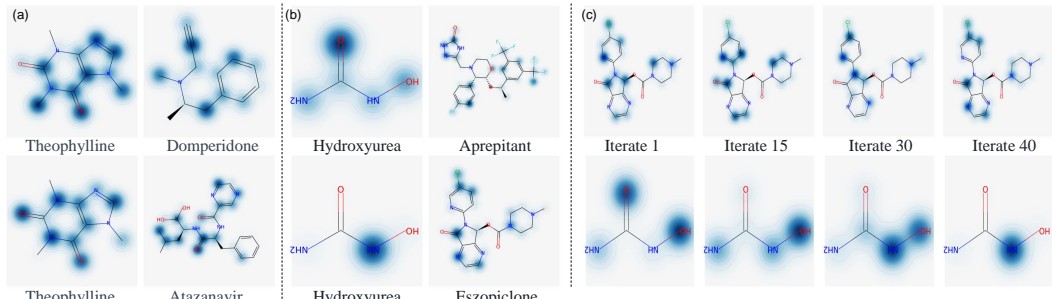

Figure 4: Interpretability analysis on interactive substructure. The visual representation of the interaction between (a) Theophylline molecules and different molecules. And (b) Hydroxyurea molecules. (c) Substructure visualization results under different iteration numbers. The darker the color means, the greater the weight.

importance. This is evident in the gradual clarification and stabilization of substructures, accompanied by the gradual removal of redundant nodes.

## 5 LIMITATION AND FUTURE OUTLOOK

ISE implements the dynamic substructure extraction process of molecular pairs based on the EM algorithm and has achieved significant improvement in the MRL experiment. However, considering the diversity and complexity of the real chemical molecular space, we expect to improve the current framework in three aspects in the future: 1) Expect to obtain more molecular interaction processes and analyses between molecules, which is limited by the limitations of current research, we only verified it during the interaction process of two molecules. However, the interaction system of multiple molecules is still a research hotspot that cannot be ignored. 2) It is expected to obtain a more efficient iterative pruning strategy. For larger data sets, ISE requires more IN times, which will undoubtedly increase the consumption of resources and time; 3) Anticipated to be effective in verifying large molecular data sets, our focus extends beyond the tested molecular interaction tasks. Interactions between macromolecules such as protein-protein, protein-peptide, and drug-protein also represent significant molecular interaction tasks. Acknowledging the differences in macromolecule modeling methods, we aim to delve into the exploration of macromolecular interactions in our future work.

## 6 CONCLUSION

This paper presents significant advancements in the field of molecular relational learning through the introduction of the ISE framework and IGIB theory. These methodologies address the crucial limitations of existing methods, particularly those pertaining to core substructure extraction. These advancements provide a much-needed paradigm shift in the understanding and analysis of molecular interactions, emphasizing the importance of the dynamic interactions between substructures. The ISE framework, firmly supported by experimental validation, has shown superiority in accuracy, generalizability, and interpretability. The framework's generalizability suggests its potential application in numerous areas, expanding the boundaries of the field. Moreover, the introduction of the IGIB theory has revitalized the interpretative study of core substructures. This theory, guided by the philosophy of effective information utilization, provides valuable insights into the selection process of essential interactive substructures. These insights facilitate a more nuanced understanding of molecular dynamics. These insights facilitate a more nuanced understanding of molecular dynamics, which has the potential to reshape our approach to molecular relational learning, stimulating more in-depth and insightful research in the future.

# 7 ACKNOWLEDGMENTS

This paper is partially supported by the National Natural Science Foundation of China (No.12227901, No.62072427), the Project of Stable Support for Youth Team in Basic Research Field, CAS (No.YSBR-005), Academic Leaders Cultivation Program, USTC.

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

# A    RELATED WORK

## A.1    MOLECULAR RELATIONAL LEARNING

Molecular research Liu et al. (2025); Zhang et al. (2025) is a critical task in the field of natural sciences, and molecular relationship learning (MRL) is a crucial branch of this area. However, it still faces a variety of challenges, including drug-drug interaction (DDI) prediction and solvation-free energy prediction for solute-solvent pairs. The rapid emergence of graph neural networks Kipf & Welling (2016); Huang et al. (2024) (GNNs) has ignited considerable interest in employing graph-based methodologies for MRL. For instance, Zhong et al.Zhong et al. (2019) harnessed Graph Convolutional Neural Networks (GCNNs) for message aggregation and utilized an attention-based pooling method to forecast DDIs. Jones et al.Zhang et al. (2022) employed a GNN to accurately predict water solvation Gibbs free energies for over 100,000 organic compounds, achieving an impressive error rate of 0.4 kcal/mol. The intricate relationship between two molecules is inevitably influenced by their specific substructures and functionalities Harrold & Zavod (2014); Fu et al. (2020).

As a consequence, research has shifted towards substructure extraction and the interplay between these substructures. Yu et al. Yu et al. (2022a) integrated functional group information of drug molecules as substructures, further exploring the interactions among them. Nyamabo et al. (2021) introduced the Substructure-Substructure Interaction for Drug-Drug Interaction (SSI-DDI) method Nyamabo et al. (2021), employing Graph Attention Network (GAT) layers for substructure extraction and co-attention layers for modeling interactions among substructures.

However, prevailing methodologies typically encode two molecules separately or extract substructures independently, thereby overlooking their interaction for specific tasks. To capture the interaction between molecules during substructure extraction, Lee et al.Lee et al. (2023a) introduced the Conditional Graph Information Bottleneck (CGIB) model. This model, inspired by Information Bottleneck theory, identifies core substructures between pairs of graphs and predicts interaction behavior. Aligned with the Structural Causal Model (SCM), Lee et al.Lee et al. (2023b) introduced a conditional intervention framework where interventions are conditioned on paired molecules. This framework enables the model to effectively glean insights from causal substructures and mitigate the confounding effects of spuriously correlated shortcut substructures in chemical reactions. Despite its demonstrated superiority over prior methods, the interaction mechanism remains rudimentary. Direct interaction between entire graphs introduces excessive redundant information, hindering the extraction of interacting substructures. This is because molecules often operate through one or several core substructures. In contrast, our method focuses on leveraging the interaction of key substructures, particularly under task-specific conditions, during the exploration process.

## A.2    GRAPH INFORMATION BOTTLENECK

The GIB theory offers a precise method for obtaining subgraphs and has been widely applied in the field of extracting subgraphs from a single graph. PGIB Yu et al. (2020) proposes a Graph Information Bottleneck (GIB) framework for recognizing informative yet compact subgraphs from the original graph, addressing key graph learning problems like graph denoising and compression. To optimize the challenging GIB objective, it introduces a mutual information estimator for irregular graph data, a bi-level optimization scheme, and a connectivity loss to stabilize the process. VGIB Yu et al. (2022b) further stabilizes the subgraph recognition process by injecting Gaussian noise into node representations, modulating the information flow from the original graph to the perturbed graph.

Additionally, Lee et al. Lee et al. (2023a) expanded the graph information bottleneck to the field of molecular relational learning, proposing the Conditional Graph Information Bottleneck (CGIB) theory, which aims to retain as much relevant information as possible with paired graphs while obtaining compressed subgraphs. The CGIB theory addresses the issue of extracting independent subgraphs in GIB for MRL tasks, but considering all information from another graph during interaction can introduce excessive noise. To address this limitation, this paper proposes the IGIB theory, which fully considers the detailed molecular interactions in molecular relational learning to ensure precise extraction of interaction subgraphs.

# B PROOFS

In this section, we provide detailed derivations for the theoretical aspects and equations presented in this paper. First, we explore the impact of redundant substructures on the extraction of core substructures in Section B.1. Additionally, we describe two Derivation Proofs of the E-step and M-step for ISE in Section B.2 and B.3. In Section B.4, we analyze the convergence of ISE. Section B.5 and B.6 focus on providing detailed proofs for the $\mathcal{L}_{pre}$ and $\mathcal{L}_{com1}$ loss formulas of the IGIB theory.

## B.1 IMPACT OF REDUNDANT SUBSTRUCTURES ON CORE SUBSTRUCTURE EXTRACTION

Let $\mathcal{G}_{s1}$ denote a general substructure of $\mathcal{G}_1$, $\mathcal{G}_{IB1}$ the core substructure of $\mathcal{G}_1$, $\mathcal{G}_{IB2}$ the core substructure of $\mathcal{G}_2$, and $\mathcal{G}_{n2}$ the redundant substructure of $\mathcal{G}_2$.

**1. Objective Function in Existing Methods:** The extraction of the core substructure $\mathcal{G}_{IB1}$ often involves minimizing mutual information. Specifically, this can be formulated as the following optimization problem:

$$\arg\min_{\mathcal{G}_{s1}} I\left(\mathcal{G}_{s1}; \mathcal{G}_1 \mid \mathcal{G}_2\right), \tag{17}$$

where $I\left(\cdot\right)$ represents the mutual information between variables.

**2. Decomposition of $\mathcal{G}_2$:** The structure $\mathcal{G}_2$ can be decomposed into two components:

- The core substructure $\mathcal{G}_{IB2}$, which contains essential and valid information.
- The redundant substructure $\mathcal{G}_{n2}$, which primarily introduces noise or irrelevant information.

Thus, $\mathcal{G}_2$ can be expressed as:

$$\mathcal{G}_2 = \mathcal{G}_{IB2} + \mathcal{G}_{n2}. \tag{18}$$

Substituting this decomposition into the objective function, we obtain:

$$\arg\min_{\mathcal{G}_{s1}} I\left(\mathcal{G}_{s1}; \mathcal{G}_1 \mid \mathcal{G}_{IB2} + \mathcal{G}_{n2}\right). \tag{19}$$

**3. Conditional Independence Analysis:** Assume that $\mathcal{G}_{n2}$ is conditionally independent of $\mathcal{G}_{IB1}$, as the redundant substructure does not directly influence the core substructure's information. Using the chain rule of mutual information, we can expand the objective function as:

$$I\left(\mathcal{G}_{s1}; \mathcal{G}_1 \mid \mathcal{G}_{IB2} + \mathcal{G}_{n2}\right) = I\left(\mathcal{G}_{s1}; \mathcal{G}_1 \mid \mathcal{G}_{IB2}\right) + I\left(\mathcal{G}_{s1}; \mathcal{G}_1 \mid \mathcal{G}_{n2}\right). \tag{20}$$

**4. Impact of Redundancy:** The term $I\left(\mathcal{G}_{s1}; \mathcal{G}_1 \mid \mathcal{G}_{n2}\right)$ captures the influence of the redundant structure $\mathcal{G}_{n2}$ on the extraction process. However, as $\mathcal{G}_{n2}$ mainly introduces noise or irrelevant information, this term interferes with the actual optimization target, leading to suboptimal results. Ideally, the objective function should exclude the redundant substructure, focusing solely on the core substructure $\mathcal{G}_{IB2}$:

$$\arg\min_{\mathcal{G}_{s1}} I\left(\mathcal{G}_{s1}; \mathcal{G}_1 \mid \mathcal{G}_{IB2}\right). \tag{21}$$

**5. Conclusion:** Incorporating the redundant substructure $\mathcal{G}_{n2}$ into the optimization process introduces an additional mutual information term, $I\left(\mathcal{G}_{s1}; \mathcal{G}_1 \mid \mathcal{G}_{n2}\right)$, which interferes with the extraction of core substructures. To achieve more accurate and efficient extraction, the optimization should rely exclusively on the core substructure $\mathcal{G}_{IB2}$, thereby eliminating the influence of redundant information.

## B.2 DERIVATION PROOF 1 OF THE E-STEP AND M-STEP FOR ISE

*Proof.* **Objective:** Given two graphs $\mathcal{G}_1$ and $\mathcal{G}_2$ and label $Y$, we aim to identify the substructures $\mathcal{G}_{s1}$ and $\mathcal{G}_{s2}$ most relevant to label $Y$. $\mathbf{Y}_{\mathcal{G}}$ is the observed variables. Additionally, $\mathbf{Y}_{\mathcal{G}}$ is determined by the model parameters $\theta$ as follows:

$$\mathbf{Y}_{\mathcal{G}} = \arg\max_{\mathbf{Y}_{\mathcal{G}}} \log P\left(\mathbf{Y}_{\mathcal{G}} \mid \theta\right). \tag{22}$$

**Latent Variables:** In molecular interactions, core substructures frequently exert significant influence. Consequently, $\mathcal{G}_2$ primarily influences through latent variables $\mathcal{G}_{s2}$, where:

$$P\left(\mathbf{Y}_{\mathcal{G}} \mid \theta\right) = \int_{\mathcal{G}_{s2}} P\left(\mathbf{Y}_{\mathcal{G}}, \mathcal{G}_{s2} \mid \theta\right) d\mathcal{G}_{s2}. \tag{23}$$

**Bayes' Theorem Application:** Applying Bayes' theorem, we derive:

$$P\left(\mathbf{Y}_{\mathcal{G}} \mid \theta\right) = \frac{P\left(\mathbf{Y}_{\mathcal{G}}, \mathcal{G}_{s2} \mid \theta\right)}{P\left(\mathcal{G}_{s2} \mid \mathbf{Y}_{\mathcal{G}}, \theta\right)}. \tag{24}$$

**Logarithmic Transformation:** Taking the logarithm on both sides and introducing the probability distribution of $\mathcal{G}_{s2}$ as $q(\mathcal{G}_{s2})$, while ensuring $\int_{\mathcal{G}_{s2}} q\left(\mathcal{G}_{s2}\right) d\mathcal{G}_{s2} = 1$, we arrive at:

$$\log P\left(\mathbf{Y}_{\mathcal{G}} \mid \theta\right) = \\ \log \frac{P\left(\mathbf{Y}_{\mathcal{G}}, \mathcal{G}_{s2} \mid \theta\right)}{q\left(\mathcal{G}_{s2}\right)} - \log \frac{P\left(\mathcal{G}_{s2} \mid \mathbf{Y}_{\mathcal{G}}, \theta\right)}{q\left(\mathcal{G}_{s2}\right)}. \tag{25}$$

**Expectation Calculation:** By taking the expectation with respect to $\mathcal{G}_{s2}$ on both sides and converting it into integral form, we obtain:

$$\int_{\mathcal{G}_{s2}} q\left(\mathcal{G}_{s2}\right) P\left(\mathbf{Y}_{\mathcal{G}} \mid \theta\right) d\mathcal{G}_{s2} = \\ \int_{\mathcal{G}_{s2}} q\left(\mathcal{G}_{s2}\right) \log \frac{P\left(\mathbf{Y}_{\mathcal{G}}, \mathcal{G}_{s2} \mid \theta\right)}{q\left(\mathcal{G}_{s2}\right)} d\mathcal{G}_{s2} \\ - \int_{\mathcal{G}_{s2}} q\left(\mathcal{G}_{s2}\right) \log \frac{P\left(\mathcal{G}_{s2} \mid \mathbf{Y}_{\mathcal{G}}, \theta\right)}{q\left(\mathcal{G}_{s2}\right)} d\mathcal{G}_{s2}. \tag{26}$$

**ELBO Derivation:** Simplifying leads to:

$$\log P\left(\mathbf{Y}_{\mathcal{G}} \mid \theta\right) = \\ \int_{\mathcal{G}_{s2}} q\left(\mathcal{G}_{s2}\right) \log \frac{P\left(\mathbf{Y}_{\mathcal{G}}, \mathcal{G}_{s2} \mid \theta\right)}{q\left(\mathcal{G}_{s2}\right)} \\ + KL\left(q\left(\mathcal{G}_{s2}\right) \| P\left(\mathcal{G}_{s2} \mid \mathbf{Y}_{\mathcal{G}}, \theta\right)\right). \tag{27}$$

**ELBO Inequality:** Due to the non-negativity of the Kullback-Leibler divergence, we establish:

$$\log P\left(\mathbf{Y}_{\mathcal{G}} \mid \theta\right) \geq \int_{\mathcal{G}_{s2}} q\left(\mathcal{G}_{s2}\right) \log \frac{P\left(\mathbf{Y}_{\mathcal{G}}, \mathcal{G}_{s2} \mid \theta\right)}{q\left(\mathcal{G}_{s2}\right)}. \tag{28}$$

**ELBO Optimization:** By setting the KL divergence term to zero, we aim to maximize the expectation, leading to:

$$\int_{\mathcal{G}_{s2}} q\left(\mathcal{G}_{s2}\right) \log \frac{P\left(\mathcal{G}_{s2}, \mathbf{Y}_{\mathcal{G}} \mid \theta\right)}{q\left(\mathcal{G}_{s2}\right)} \\ = \int_{\mathcal{G}_{s2}} P\left(\mathcal{G}_{s2} \mid \mathbf{Y}_{\mathcal{G}}, \theta\right) \log \frac{P\left(\mathcal{G}_{s2}, \mathbf{Y}_{\mathcal{G}} \mid \theta\right)}{P\left(\mathcal{G}_{s2} \mid \theta, \mathbf{Y}_{\mathcal{G}}\right)} d\mathcal{G}_{s2} \\ = \mathbb{E}_{\mathcal{G}_{s2} \mid \mathbf{Y}_{\mathcal{G}}, \theta}\left[\log \frac{P\left(\mathcal{G}_{s2}, \mathbf{Y}_{\mathcal{G}} \mid \theta\right)}{P\left(\mathcal{G}_{s2} \mid \theta, \mathbf{Y}_{\mathcal{G}}\right)}\right] \\ = \mathbb{E}_{\mathcal{G}_{s2} \mid \theta, \mathbf{Y}_{\mathcal{G}}}\left[\log P\left(\mathcal{G}_{s2}, \mathbf{Y}_{\mathcal{G}} \mid \theta\right)\right] - \\ \mathbb{E}_{\mathcal{G}_{s2} \mid \theta, \mathbf{Y}_{\mathcal{G}}}\left[\log P\left(\mathcal{G}_{s2} \mid \theta, \mathbf{Y}_{\mathcal{G}}\right)\right]. \tag{29}$$

We proceed by meticulously examining the model parameters $\theta$, with a particular focus on its components within the MRL task. The parameter set $\theta$ consists of only two constituents, $\theta_1$ and $\theta_2$. In this context, $\theta_1$ governs the selection of $\mathcal{G}_{s1}$ from the set $\mathcal{G}_1$, while $\theta_2$ regulates intermolecular interactions to determine $\mathcal{G}_{s2}$ based on $\mathcal{G}_{s1}$. Notably, the supervision of molecular interaction,

indicated by the label $Y$, is independent of the EM iteration process, rendering it negligible for our analysis. Consequently, our focus narrows down to $\theta_1$, which exclusively influences $\mathcal{G}_{s1}$. However, concerning $\mathcal{G}_{s2}$, $\mathcal{G}_{s1}$ emerges as the determinant parameter. Thus, we redefine $\theta$ as $\mathcal{G}_{s1}$. Consequently, at the current time step $(t)$, the E-step can be succinctly expressed as follows:

$$\mathbb{E}_{\mathcal{G}_{s2}^{(t)}|\mathbf{Y}_{\mathcal{G}},\mathcal{G}_{s1}^{(t)}}\left[\log\frac{P\left(\mathcal{G}_{s2}^{(t)},\mathbf{Y}_{\mathcal{G}}\mid\mathcal{G}_{s1}^{(t)}\right)}{P\left(\mathcal{G}_{s2}^{(t)}\mid\mathbf{Y}_{\mathcal{G}},\mathcal{G}_{s1}^{(t)}\right)}\right]. \tag{30}$$

Subsequently, we aim to maximize this expectation, leading to:

$$\mathcal{G}_{s1}^{(t+1)}=\arg\max_{\mathcal{G}_{s1}}\mathbb{E}_{\mathcal{G}_{s2}^{(t)}|\mathbf{Y}_{\mathcal{G}},\mathcal{G}_{s1}^{(t)}}\left[\log\frac{P\left(\mathcal{G}_{s2}^{(t)},\mathbf{Y}_{\mathcal{G}}\mid\mathcal{G}_{s1}^{(t)}\right)}{P\left(\mathcal{G}_{s2}^{(t)}\mid\mathbf{Y}_{\mathcal{G}},\mathcal{G}_{s1}^{(t)}\right)}\right]. \tag{31}$$

$\square$

### B.3 DERIVATION PROOF 2 OF THE E-STEP AND M-STEP FOR ISE

*Proof.* **Introduction of Latent Variables:** We introduce latent variables using the Law of Total Probability:

$$\log P(\mathbf{Y}_{\mathcal{G}}\mid\mathcal{G}_{s1})=\log\int_{\mathcal{G}_{s2}}P(\mathbf{Y}_{\mathcal{G}},\mathcal{G}_{s2}\mid\mathcal{G}_{s1})d\mathcal{G}_{s2} \tag{32}$$

**Incorporating Probability Distribution:** Assuming the probability distribution of $\mathcal{G}_{s2}$ as $q(\mathcal{G}_{s2})$, we rewrite the equation as:

$$\log P(\mathbf{Y}_{\mathcal{G}}\mid\mathcal{G}_{s1})=\log\int_{\mathcal{G}_{s2}}\frac{P(\mathbf{Y}_{\mathcal{G}},\mathcal{G}_{s2}\mid\mathcal{G}_{s1})}{q(\mathcal{G}_{s2})}q(\mathcal{G}_{s2})d\mathcal{G}_{s2} \tag{33}$$

**Expectation Calculation:** The integral inside the log is the expectation of $\frac{P(\mathbf{Y}_{\mathcal{G}},\mathcal{G}_{s2}|\mathcal{G}_{s1})}{q(\mathcal{G}_{s2})}$ with respect to $\mathcal{G}_{s2}$:

$$\log P(\mathbf{Y}_{\mathcal{G}}\mid\mathcal{G}_{s1})=\log\mathbb{E}_{\mathcal{G}_{s2}}\left[\frac{P(\mathbf{Y}_{\mathcal{G}},\mathcal{G}_{s2}\mid\mathcal{G}_{s1})}{q(\mathcal{G}_{s2})}\right] \tag{34}$$

**Jensen's Inequality Application:** Since the logarithm function is strictly concave, according to Jensen's inequality:

$$\log\mathbb{E}_{\mathcal{G}_{s2}}\left[\frac{P(\mathbf{Y}_{\mathcal{G}},\mathcal{G}_{s2}\mid\mathcal{G}_{s1})}{q(\mathcal{G}_{s2})}\right]\geq\mathbb{E}_{\mathcal{G}_{s2}}\left[\log\frac{P(\mathbf{Y}_{\mathcal{G}},\mathcal{G}_{s2}\mid\mathcal{G}_{s1})}{q(\mathcal{G}_{s2})}\right] \tag{35}$$

**ELBO Derivation:** This implies:

$$\log P(\mathbf{Y}_{\mathcal{G}}\mid\mathcal{G}_{s1})\geq\mathbb{E}_{\mathcal{G}_{s2}}\left[\log\frac{P(\mathbf{Y}_{\mathcal{G}},\mathcal{G}_{s2}\mid\mathcal{G}_{s1})}{q(\mathcal{G}_{s2})}\right]=\int_{\mathcal{G}_{s2}}q(\mathcal{G}_{s2})\log\frac{P(\mathbf{Y}_{\mathcal{G}},\mathcal{G}_{s2}\mid\mathcal{G}_{s1})}{q(\mathcal{G}_{s2})}d\mathcal{G}_{s2} \tag{36}$$

**Equality Condition of Jensen's Inequality:** The equality holds when $\frac{P(\mathbf{Y}_{\mathcal{G}},\mathcal{G}_{s2}|\mathcal{G}_{s1})}{q(\mathcal{G}_{s2})}=C$:

$$q(\mathcal{G}_{s2})=\frac{1}{C}P(\mathbf{Y}_{\mathcal{G}},\mathcal{G}_{s2}\mid\mathcal{G}_{s1}) \tag{37}$$

**Normalization:** Integrating both sides with respect to $\mathcal{G}_{s2}$:

$$\int_{\mathcal{G}_{s2}}q(\mathcal{G}_{s2})d\mathcal{G}_{s2}=1\implies\frac{1}{C}P(\mathbf{Y}_{\mathcal{G}}\mid\mathcal{G}_{s1})=1 \tag{38}$$

**Final Inference:** Hence, $P(\mathbf{Y}_{\mathcal{G}}\mid\mathcal{G}_{s1})=C$. Replacing $C$:

$$q(\mathcal{G}_{s2})=\frac{P(\mathbf{Y}_{\mathcal{G}},\mathcal{G}_{s2}\mid\mathcal{G}_{s1})}{P(\mathbf{Y}_{\mathcal{G}}\mid\mathcal{G}_{s1})} \tag{39}$$

**ELBO Formulation:** Clearly, this expression is the Evidence Lower Bound (ELBO) we mentioned earlier.

$$\int_{\mathcal{G}_{s2}} P(\mathcal{G}_{s2} \mid \mathbf{Y}_{\mathcal{G}}, \mathcal{G}_{s1}) \log \frac{P(\mathbf{Y}_{\mathcal{G}}, \mathcal{G}_{s2} \mid \mathcal{G}_{s1})}{P(\mathcal{G}_{s2} \mid \mathbf{Y}_{\mathcal{G}}, \mathcal{G}_{s1})} d\mathcal{G}_{s2} = \mathbb{E}_{\mathcal{G}_{s2} \mid \mathbf{Y}_{\mathcal{G}}, \mathcal{G}_{s1}} \left[ \log \frac{P(\mathbf{Y}_{\mathcal{G}}, \mathcal{G}_{s2} \mid \mathcal{G}_{s1})}{P(\mathcal{G}_{s2} \mid \mathbf{Y}_{\mathcal{G}}, \mathcal{G}_{s1})} \right] \quad (40)$$

**Iterative Procedure:** Since the ISE framework is iterative, in each iteration, we first estimate the posterior $P\left( \mathcal{G}_{s2}^{(t)} \mid \mathbf{Y}_{\mathcal{G}}, \mathcal{G}_{s1}^{(t)} \right)$ based on the previous iteration's $\mathcal{G}_{s1}^{(t)}$ and the samples $\mathbf{Y}_{\mathcal{G}}$, and then compute the expectation in the E-step:

$$\mathbb{E}_{\mathcal{G}_{s2}^{(t)} \mid \mathbf{Y}_{\mathcal{G}}, \mathcal{G}_{s1}^{(t)}} \left[ \log \frac{P\left( \mathcal{G}_{s2}^{(t)}, \mathbf{Y}_{\mathcal{G}} \mid \mathcal{G}_{s1}^{(t)} \right)}{P\left( \mathcal{G}_{s2}^{(t)} \mid \mathbf{Y}_{\mathcal{G}}, \mathcal{G}_{s1}^{(t)} \right)} \right]. \quad (41)$$

**Maximization Objective:** Subsequently, we aim to maximize this expectation, leading to:

$$\mathcal{G}_{s1}^{(t+1)} = \arg\max_{\mathcal{G}_{s1}} \mathbb{E}_{\mathcal{G}_{s2}^{(t)} \mid \mathbf{Y}_{\mathcal{G}}, \mathcal{G}_{s1}^{(t)}} \left[ \log \frac{P\left( \mathcal{G}_{s2}^{(t)}, \mathbf{Y}_{\mathcal{G}} \mid \mathcal{G}_{s1}^{(t)} \right)}{P\left( \mathcal{G}_{s2}^{(t)} \mid \mathbf{Y}_{\mathcal{G}}, \mathcal{G}_{s1}^{(t)} \right)} \right]. \quad (42)$$

$\square$

### B.4 PROOF OF CONVERGENCE OF THE EM ALGORITHM

*Proof.* The objective of the EM algorithm is to find suitable model parameters $\mathcal{G}_{s1}$ such that $P(\mathbf{Y}_{\mathcal{G}} \mid \mathcal{G}_{s1})$ is maximized. Since EM is an iterative algorithm, to prove its convergence, it suffices to show that $P\left( \mathbf{Y}_{\mathcal{G}} \mid \mathcal{G}_{s1}^{(t+1)} \right) \geq P\left( \mathbf{Y}_{\mathcal{G}} \mid \mathcal{G}_{s1}^{(t)} \right)$ holds.

As we know:

$$\log P(\mathbf{Y}_{\mathcal{G}} \mid \mathcal{G}_{s1}) = \log P(\mathbf{Y}_{\mathcal{G}}, \mathcal{G}_{s2} \mid \mathcal{G}_{s1}) - \log P(\mathcal{G}_{s2} \mid \mathbf{Y}_{\mathcal{G}}, \mathcal{G}_{s1}) \quad (43)$$

Taking the expectation with respect to $(\mathcal{G}_{s2} \mid \mathbf{Y}_{\mathcal{G}}, \mathcal{G}_{s1}^{(t)})$ on both sides, we have:

$$\mathbb{E}_{\mathcal{G}_{s2} \mid \mathbf{Y}_{\mathcal{G}}, \mathcal{G}_{s1}^{(t)}} [\log P(\mathbf{Y}_{\mathcal{G}} \mid \mathcal{G}_{s1})]$$
$$= \mathbb{E}_{\mathcal{G}_{s2} \mid \mathbf{Y}_{\mathcal{G}}, \mathcal{G}_{s1}^{(t)}} [\log P(\mathbf{Y}_{\mathcal{G}}, \mathcal{G}_{s2} \mid \mathcal{G}_{s1})] - \mathbb{E}_{\mathcal{G}_{s2}^{(t)} \mid \mathbf{Y}_{\mathcal{G}}, \mathcal{G}_{s1}^{(t)}} [\log P(\mathcal{G}_{s2} \mid \mathbf{Y}_{\mathcal{G}}, \mathcal{G}_{s1})] \quad (44)$$

For the left-hand side of the equation, since $\log P(\mathbf{Y}_{\mathcal{G}} \mid \mathcal{G}_{s1})$ is independent of $\mathcal{G}_{s2}$, we have:

$$\mathbb{E}_{\mathcal{G}_{s2} \mid \mathbf{Y}_{\mathcal{G}}, \mathcal{G}_{s1}^{(t)}} [\log P(\mathbf{Y}_{\mathcal{G}} \mid \mathcal{G}_{s1})]$$
$$= \int_{\mathcal{G}_{s2}} P\left( \mathcal{G}_{s2} \mid \mathbf{Y}_{\mathcal{G}}, \mathcal{G}_{s1}^{(t)} \right) \log P(\mathbf{Y}_{\mathcal{G}} \mid \mathcal{G}_{s1}) d\mathcal{G}_{s2} = \log P(\mathbf{Y}_{\mathcal{G}} \mid \mathcal{G}_{s1}) \quad (45)$$

For the right-hand side, let's first consider the first term. It is actually the $Q\left( \mathcal{G}_{s1}, \mathcal{G}_{s1}^{(t)} \right)$ from our E-step, as:

$$\mathcal{G}_{s1}^{(t+1)} = \arg\max_{\mathcal{G}_{s1}} Q\left( \mathcal{G}_{s1}, \mathcal{G}_{s1}^{(t)} \right) \quad (46)$$

Therefore, it follows that:

$$Q\left( \mathcal{G}_{s1}^{(t+1)}, \mathcal{G}_{s1}^{(t)} \right) \geq Q\left( \mathcal{G}_{s1}, \mathcal{G}_{s1}^{(t)} \right) \quad (47)$$

Since $\mathcal{G}_{s1}$ is a variable, we can set $\mathcal{G}_{s1} = \mathcal{G}_{s1}^{(t)}$, thus:

$$Q\left( \mathcal{G}_{s1}^{(t+1)}, \mathcal{G}_{s1}^{(t)} \right) \geq Q\left( \mathcal{G}_{s1}^{(t)}, \mathcal{G}_{s1}^{(t)} \right) \quad (48)$$

Next, let's consider the second term. Since we aim to prove $\log P\left(\mathbf{Y}_\mathcal{G} \mid \mathcal{G}_{s1}^{(t+1)}\right) \geq \log P\left(\mathbf{Y}_\mathcal{G} \mid \mathcal{G}_{s1}^{(t)}\right)$, and we have already demonstrated $Q\left(\mathcal{G}_{s1}^{(t+1)}, \mathcal{G}_{s1}^{(t)}\right) \geq Q\left(\mathcal{G}_{s1}^{(t)}, \mathcal{G}_{s1}^{(t)}\right)$, we only need to ensure that the second term satisfies

$$\mathbb{E}_{\mathcal{G}_{s2}^{(t)} \mid \mathbf{Y}_\mathcal{G}, \mathcal{G}_{s1}^{(t)}} \left[\log P\left(\mathcal{G}_{s2}^{(t)} \mid \mathbf{Y}_\mathcal{G}, \mathcal{G}_{s1}^{(t+1)}\right)\right]$$

$$\leq \mathbb{E}_{\mathcal{G}_{s2}^{(t)} \mid \mathbf{Y}_\mathcal{G}, \mathcal{G}_{s1}^{(t)}} \left[\log P\left(\mathcal{G}_{s2}^{(t)} \mid \mathbf{Y}_\mathcal{G}, \mathcal{G}_{s1}^{(t)}\right)\right] \tag{49}$$

$$\leq 0$$

Thus, it is proven

$$\log P\left(\mathbf{Y}_\mathcal{G} \mid \mathcal{G}_{s1}^{(t+1)}\right) \geq \log P\left(\mathbf{Y}_\mathcal{G} \mid \mathcal{G}_{s1}^{(t)}\right) \tag{50}$$

$\square$

## B.5 Proof of $\mathcal{L}_{pre}$

*Proof.* Regarding $I\left(\mathbf{Y}; \mathcal{G}_{s1}, \mathcal{G}_{s2}\right)$, we consider $P_\theta\left(\mathbf{Y} \mid \mathcal{G}_{s1}, \mathcal{G}_{s2}\right)$ as the variational estimation of $P\left(\mathbf{Y} \mid \mathcal{G}_{s1}, \mathcal{G}_{s2}\right)$. Therefore, we can proceed with the following derivation:

$$I\left(\mathbf{Y}; \mathcal{G}_{s1}, \mathcal{G}_{s2}\right) = \mathbb{E}_{(\mathbf{Y}, \mathcal{G}_{s1}, \mathcal{G}_{s2})} \log\left[\frac{P\left(\mathbf{Y} \mid \mathcal{G}_{s1}, \mathcal{G}_{s2}\right)}{P(\mathbf{Y})}\right]$$

$$= \mathbb{E}_{(\mathbf{Y}, \mathcal{G}_{s1}, \mathcal{G}_{s2})} \log\left[\frac{P_\theta\left(\mathbf{Y} \mid \mathcal{G}_{s1}, \mathcal{G}_{s2}\right)}{P(\mathbf{Y})}\right] + \tag{51}$$

$$\mathbb{E}_{(\mathcal{G}_{s1}, \mathcal{G}_{s2})} \log\left[KL\left(P\left(\mathbf{Y} \mid \mathcal{G}_{s1}, \mathcal{G}_{s2}\right) \| P_\theta\left(\mathbf{Y} \mid \mathcal{G}_{s1}, \mathcal{G}_{s2}\right)\right)\right].$$

Considering the non-negativity property of the Kullback-Leibler divergence, we can conclude that:

$$I\left(\mathbf{Y}; \mathcal{G}_{s1}, \mathcal{G}_{s2}\right) \geq \mathbb{E}_{(\mathbf{Y}, \mathcal{G}_{s1}, \mathcal{G}_{s2})} \log\left[\frac{P_\theta\left(\mathbf{Y} \mid \mathcal{G}_{s1}, \mathcal{G}_{s2}\right)}{P(\mathbf{Y})}\right]$$

$$= \mathbb{E}_{(\mathbf{Y}, \mathcal{G}_{s1}, \mathcal{G}_{s2})} \log\left[P_\theta\left(\mathbf{Y} \mid \mathcal{G}_{s1}, \mathcal{G}_{s2}\right)\right] + H(\mathbf{Y}). \tag{52}$$

As $H(\mathbf{Y})$ remains constant across all data, it can be omitted, resulting in the final formulation of this term:

$$\mathcal{L}_{pre} := \mathbb{E}_{(\mathbf{Y}, \mathcal{G}_{s1}, \mathcal{G}_{s2})} \log\left[P_\theta\left(\mathbf{Y} \mid \mathcal{G}_{s1}, \mathcal{G}_{s2}\right)\right]. \tag{53}$$

$\square$

## B.6 Proof of $\mathcal{L}_{com1}$

*Proof.* For $I\left(\mathcal{G}_{s1}; \mathcal{G}_1, \mathcal{G}_{s2}\right)$, $z_{\mathcal{G}_{s1}}$ is employed to denote the encoding of $\mathcal{G}_{s1}$, and we approximate $I\left(z_{\mathcal{G}_{s1}}; \mathcal{G}_1, \mathcal{G}_{s2}\right)$ using a variational inference approach $q(z_{\mathcal{G}_{s1}})$ as an estimate for $p(z_{\mathcal{G}_{s1}})$:

$$I\left(z_{\mathcal{G}_{s1}}; \mathcal{G}_1, \mathcal{G}_{s2}\right) = \mathbb{E}_{\left(z_{\mathcal{G}_{s1}}, \mathcal{G}_1, \mathcal{G}_{s2}\right)} \log\left[\frac{p_\Phi\left(z_{\mathcal{G}_{s1}} \mid \mathcal{G}_1, \mathcal{G}_{s2}\right)}{p\left(z_{\mathcal{G}_{s1}}\right)}\right]$$

$$= \mathbb{E}_{(\mathcal{G}_1, \mathcal{G}_{s2})} \log\left[\frac{p_\Phi\left(z_{\mathcal{G}_{s1}} \mid \mathcal{G}_1, \mathcal{G}_{s2}\right)}{q\left(z_{\mathcal{G}_{s1}}\right)}\right] - \tag{54}$$

$$\mathbb{E}_{\left(z_{\mathcal{G}_{s1}}, \mathcal{G}_1, \mathcal{G}_{s2}\right)} KL\left(p\left(z_{\mathcal{G}_{s1}}\right) \| q\left(z_{\mathcal{G}_{s1}}\right)\right).$$

With the non-negativity property of the Kullback-Leibler divergence, we can conclude that:

$$I\left(z_{\mathcal{G}_{s1}}; \mathcal{G}_1, \mathcal{G}_{s2}\right) \leq$$

$$\mathbb{E}_{(\mathcal{G}_1, \mathcal{G}_{s2})} KL\left(p_\Phi\left(z_{\mathcal{G}_{s1}} \mid \mathcal{G}_1, \mathcal{G}_{s2}\right) \| q\left(z_{\mathcal{G}_{s1}}\right)\right) := \mathcal{L}_{com1}. \tag{55}$$

Adopting the VIB framework, we postulate that the latent representation $q(z_{\mathcal{G}_{s1}})$ is derived by aggregating node representations within a fully perturbed graph. The perturbation is introduced

through noise $\epsilon$, which follows a Gaussian distribution $\mathcal{N}(\mu_{H_1}, \sigma^2_{H_1})$. The parameters $\mu_{H_1}$ and $\sigma^2_{H_1}$ represent the mean and variance of $H_1$, encapsulating information from both $\mathcal{G}_1$ and $\mathcal{G}_2$. By selecting sum pooling as the aggregation mechanism, and considering the additive property of Gaussian distributions, we formulate the following relationship:

$$q(z_{\mathcal{G}_{s1}}) = \mathcal{N}(N^1 \mu_{H_1}, N^1 \sigma^2_{H_1}), \tag{56}$$

where $N^1$ denote the number of nodes in $\mathcal{G}_1$. Then for $p_\Phi(z_{\mathcal{G}_{s1}} \mid \mathcal{G}_1, \mathcal{G}_{s2})$, we have the following equation:

$$p_\Phi(z_{\mathcal{G}_{s1}} \mid \mathcal{G}_1, \mathcal{G}_{s2}) = \mathcal{N}\left(N^1 \mu_{H_1} + \sum_{j=1}^{N^1} \lambda_j H_{1j} - \sum_{j=1}^{N^1} \lambda_j \mu_{H_1}, \sum_{j=1}^{N^1}(1-\lambda_j)^2 \sigma^2_{H_1}\right). \tag{57}$$

Finally, we have following inequality by plugging Equation equation 56 and Equation equation 57 into Equation equation 55:

$$I(z_{\mathcal{G}_{s1}}; \mathcal{G}_1, \mathcal{G}_{s2}) \leq \mathbb{E}_{\mathcal{G}_1, \mathcal{G}_{s2}}\left[-\frac{1}{2}\log A + \frac{1}{2N^1}A + \frac{B^2}{2N^1}\right] + C \tag{58}$$

where $A = \sum_{j=1}^{N^1}(1-\lambda_j)^2$, $B = \frac{\sum_{j=1}^{N^1} \lambda_j(H_{1j} - \mu_{H_1})}{\sigma_{H_1}}$ and $C$ is a constant term that is ignored during optimization. $\qquad\square$

## C FEATURES OF MOLECULAR MODELING

As illustrated in Table 3, our study leverages a carefully curated set of atomic and bond features.

Table 3: Atomic and bond features used in our study.

| Atomic Features | Bond Features |
|---|---|
| Atomic number | Bond type |
| Degree (number of bonds) | Conjugated status |
| Formal charge | Ring status |
| Chiral tag | Stereo-chemistry |
| Number of bonded H atoms | – |
| Hybridization type | – |
| Aromatic status | – |
| Mass (scaled by 0.01) | – |

## D EXPERIMENTAL SETTINGS

In this section, we provide a detailed explanation of our experimental setup. Section D.1 offers information on all the datasets used in the experiments. Section D.2 presents a basic introduction to the baselines involved in our paper. Section D.3 describes the various hyperparameters used in the model's network architecture and illustrates the hyperparameter search space and the optimal hyperparameters.

### D.1 DATASETS

**molecular interaction prediction task:** For the molecular interaction prediction task, the datasets concerning solvation free energies include MNSol, FreeSolv, CompSol, Abraham, and CombiSolv. In addition, we also selected the chromophore dataset:

- **MNSol** Marenich et al. (2020) features 3,037 experimental free energies of solvation or transfer energies across 790 unique solutes and 92 solvents. We analyze 2,275 combinations following previous work Lee et al. (2023a).

- **FreeSolv** Mobley & Guthrie (2014) offers 560 experimental and calculated hydration free energies of small molecules in water.

- **CompSol** Moine et al. (2017) explores how solvation energies are influenced by hydrogen-bonding association effects. It comprises 3,548 combinations encompassing 442 unique solutes and 259 solvents.

- **Abraham** Grubbs et al. (2010) presents 6,091 combinations featuring 1,038 unique solutes and 122 solvents.

- **CombiSolv** Vermeire & Green (2021a) amalgamates data from FreeSolv, CompSol, and Abraham, totaling 8,780 combinations.

- **Chromophore** Joung et al. (2020) encompasses 20,236 combinations of 7,016 chromophores and 365 solvents in SMILES string format. We predict key optical properties like maximum absorption wavelength (Absorption), maximum emission wavelength (Emission), and excited state lifetime (Lifetime), taking care to filter out NaN values. Due to its skewed distribution, we opt for log-normalized target values for Lifetime following previous work Lee et al. (2023a).

**drug-drug interaction prediction task:** For the drug-drug interaction prediction task, we selected the drug-drug interaction dataset. We employ positive drug pairs from MIRACLE Wang et al. (2021) and generate negative counterparts through complementary pair sampling. Detailed descriptions of each dataset are as follows:

- **ZhangDDI** Zhang et al. (2017) contains 113,972 pairwise interaction data points with 544 unique drugs.

- **ChChMiner** Zitnik et al. (2018) includes 33,669 pairwise interaction data points with 997 unique drugs.

- **DeepDDI** Ryu et al. (2018) encompasses 316,595 pairwise interaction data points with 1,704 unique drugs.

## D.2 BASELINES

We present a concise overview of the foundational models discussed in the experimental section, categorizing them based on the nature of the molecular interactions they consider. Category I includes models where substructure extraction is completed before any interactions occur. Some of these models capture atomic-level interactions, treating individual atoms as distinct substructures. This category emphasizes atomic interactions and ensures that the encoding of substructures remains invariant to molecular interactions. Category II comprises models where substructure extraction is influenced by the holistic graph representation of another molecule, highlighting the broader molecular context in which interactions occur. Category III encompasses models that consider the influence exerted by the substructure of another molecule on substructure extraction. This nuanced approach accounts for specific structural features within molecular interactions. For the task of drug-drug interaction prediction, we adopt the following baseline models:

**GoGNN.** Wang et al. (2020) GoGNN extracts features from structured entity graphs and entity interaction graphs in a hierarchical manner. It propose a dual attention mechanism that enables the model to preserve the importance of neighbors at both levels of the graph.

**MHCADDI.** Deac et al. (2019) A gated information transfer neural network is used to control the extraction of substructures and then interact based on an attention mechanism.

**SSI-DDI.** Nyamabo et al. (2021) It uses a 4-layer GAT network to extract substructures at different levels, and finally completes the final prediction based on the co-attention mechanism.

**CGIB.** Lee et al. (2023a) Based on the graph conditional information bottleneck theory, conditional substructures are extracted to complete the interaction between molecules.

**CMRL.** Lee et al. (2023b) CMRL detects the core substructure that is causally related to chemical reactions. It introduce a novel conditional intervention framework whose intervention is conditioned on the paired molecule.

**DSN-DDI.** Li et al. (2023) It employs local and global representation learning modules iteratively and learns drug substructures from the single drug ('intra-view') and the drug pair ('inter-view') simultaneously.

**CIGIN.** Pathak et al. (2020) is a method based on graph neural networks. The proposed model adopts an end-to-end framework consisting of three essential phases: message passing, interaction, and prediction. In the final phase, these stages are leveraged to predict solvation free energies.

For the molecular interaction prediction task, we additionally employ the following baselines:

**D-MPNN** Vermeire & Green (2021a) employs a transfer learning approach to predict solvation free energies, integrating quantum calculation fundamentals with the heightened accuracy of experimental measurements through two new databases, CombiSolv-QM and CombiSolv-Exp.

**Explainable GNN** Low et al. (2022) introduces a graph neural network (GNN) for predicting $\Delta G_{solv}$. It incorporates atom and bond-level features, semi-empirical partial atomic charges, and solvent dielectric constant into the featurization process. Solute-solvent interactions are visualized through an interaction map layer, enabling the examination of solubility-enhancing or -decreasing interactions.

**Uni-Mol** Zhou et al. (2023) incorporates two pre-trained models featuring the SE(3) Transformer architecture: a molecular model pre-trained on 209 million molecular conformations and a pocket model pre-trained on 3 million candidate protein pocket data. Additionally, Uni-Mol integrates various fine-tuning strategies to effectively apply these pre-trained models across diverse downstream tasks.

### D.3 PARAMETER SETTING

**Model Architecture.** For the molecular interaction prediction task, in the molecular coding layer, we configured the GIN network Xu et al. (2018) with 3 layers. Since the two molecules are asymmetric, each molecule has its own graph encoder and readout network. However, for DDI tasks, where the two molecules are symmetric, they share the same graph encoder and readout network.

**Model Training.** We employed the Adam optimizer for model optimization in both molecular interaction prediction and drug-drug interaction prediction task. For drug-drug interaction tasks, the learning rate was decreased by a factor of $10^{-1}$ after 10 epochs of reaching a plateau, and training was terminated when the optimal accuracy on the validation set remained unchanged for 20 consecutive epochs. Similarly, for the molecular interaction prediction task, we adopted a comparable strategy: the learning rate was reduced by a factor of $10^{-1}$ after 10 epochs of reaching a plateau, and training concluded when the optimal accuracy on the validation set did not change for 50 consecutive epochs. For the DDI task, we divided the dataset into training, validation, and test sets in a 6:2:2 ratio. For the molecular interaction prediction task, we employed 10-fold cross-validation to partition the dataset. Our model and all baselines used the same random seed and were evaluated across five random experiments.

**Hyperparameter Tuning.** To ensure fair comparisons, we adhered to the embedding dimensions and batch sizes of the state-of-the-art baseline for each task. Detailed hyperparameter specifications are provided in Table 4. For our model, hyperparameters were fine-tuned within specified ranges: learning rate $\eta$ in $[5e^{-3}, 1e^{-3}, 5e^{-4}, 1e^{-4}, 5e^{-5}]$, $\beta_1$ in $[1e^{-1}, 1e^{-2}, 1e^{-3}, 1e^{-4}, 1e^{-6}, 1e^{-8}, 1e^{-10}]$, $\beta_2$ in $[1e^{-1}, 1e^{-2}, 1e^{-3}, 1e^{-4}, 1e^{-6}, 1e^{-8}, 1e^{-10}]$, $\tau$ in $[1.0, 0.5, 0.2]$, and $IN$ in $[3, 5, 10, 20, 30, 40, 50, 60]$.

The model implementation was conducted in PyTorch, and the execution was performed on hardware consisting of an Intel(R) Xeon(R) CPU E5-2620 v4 @ 2.10GHz and Nvidia Tesla A100 40GB. This robust hardware configuration ensures efficient processing and execution of the model.

## E ADDITIONAL EXPERIMENTS

In this section, we carry out additional experiments to demonstrate the effectiveness and interpretability of our method. In Section 7, we validate the superiority of our model using two additional classification metrics. In Section E.2, we conduct supplementary ablation experiments to gain a more comprehensive and clear understanding of the loss function derived based on the IGIB theory.

Table 4: Hyperparameter specifications ($*$: Inductive Setting 1 and $**$: Inductive Setting 2).

| | Embedding Dim ($d$) | Batch Size ($K$) | Epochs | lr | $\beta_1$ | $IN$ | $\beta_2$ | $\tau$ |
|---|---|---|---|---|---|---|---|---|
| Absorption | 52 | 256 | 500 | 5e-3 | 1e-2 | 20 | 1e-2 | 1.0 |
| Emission | 52 | 256 | 500 | 5e-3 | 1e-3 | 20 | 1e-3 | 1.0 |
| Lifetime | 52 | 256 | 500 | 5e-3 | 1e-7 | 20 | 1e-7 | 1.0 |
| MNSol | 52 | 32 | 200 | 1e-3 | 1e-7 | 10 | 1e-7 | 1.0 |
| FreeSolv | 52 | 32 | 200 | 1e-3 | 1e-9 | 3 | 1e-9 | 1.0 |
| CompSol | 52 | 256 | 500 | 1e-3 | 1e-7 | 10 | 1e-7 | 1.0 |
| Abraham | 52 | 256 | 500 | 1e-3 | 1e-11 | 5 | 1e-11 | 1.0 |
| CombiSolv | 52 | 256 | 500 | 1e-3 | 1e-7 | 10 | 1e-7 | 0.5 |
| ZhangDDI | 300 | 512 | 500 | 1e-4 | 1e-4 | 40 | 1e-4 | 0.5 |
| ChChMiner | 300 | 512 | 500 | 1e-4 | 1e-4 | 30 | 1e-4 | 0.2 |
| DeepDDI | 300 | 512 | 500 | 1e-5 | 1e-8 | 50 | 1e-8 | 1.0 |
| ZhangDDI$^*$ | 300 | 512 | 500 | 5e-5 | 1e-4 | 30 | 1e-4 | 1.0 |
| ChChMiner$^*$ | 300 | 512 | 500 | 5e-4 | 1e-4 | 20 | 1e-4 | 1.0 |
| DeepDDI$^*$ | 300 | 512 | 500 | 5e-5 | 1e-8 | 40 | 1e-8 | 1.0 |
| ZhangDDI$^{**}$ | 300 | 512 | 500 | 5e-5 | 1e-4 | 20 | 1e-4 | 1.0 |
| ChChMiner$^{**}$ | 300 | 512 | 500 | 5e-4 | 1e-4 | 20 | 1e-4 | 1.0 |
| DeepDDI$^{**}$ | 300 | 512 | 500 | 5e-5 | 1e-8 | 20 | 1e-8 | 1.0 |

In Section E.6, we provide a more detailed illustration of the dynamic process of selecting core substructures.

## E.1 THE PERFORMANCE OF OUR MODEL ON ADDITIONAL METRICS FOR THE DDI TASK.

We conducted additional experiments comparing our model with the baselines using the AUROC and F1 Score metrics. As demonstrated in Table 5 and Table 6, IGIB-ISE achieved superior results compared to the other baselines.

**AUROC (Area Under the Receiver Operating Characteristic Curve)**: AUROC measures a binary classifier's ability to distinguish between classes across all threshold values, with higher values indicating better performance.

**F1 Score**: The F1 Score is the harmonic mean of precision and recall, providing a single metric that balances both false positives and false negatives.

Table 5: Performance of different methods in transductive setting (Bold numbers are the best results).

| Method | DeepDDI | | ZhangDDI | | ChchMiner | |
|---|---|---|---|---|---|---|
| | AUROC | F1 | AUROC | F1 | AUROC | F1 |
| *Category I* | | | | | | |
| GoGNN | $92.71_{(0.27)}$ | $89.83_{(0.41)}$ | $92.35_{(0.48)}$ | $81.54_{(0.42)}$ | $96.64_{(0.40)}$ | $82.35_{(0.34)}$ |
| CIGIN | $95.35_{(0.41)}$ | $91.32_{(0.32)}$ | $91.47_{(0.55)}$ | $82.68_{(0.37)}$ | $97.29_{(0.33)}$ | $89.37_{(0.26)}$ |
| SSI-DDI | $97.42_{(0.31)}$ | $95.41_{(0.19)}$ | $93.76_{(0.34)}$ | $82.99_{(0.30)}$ | $97.81_{(0.22)}$ | $93.11_{(0.19)}$ |
| MHCADDI | $88.64_{(0.83)}$ | $88.54_{(0.55)}$ | $86.94_{(0.68)}$ | $73.67_{(0.48)}$ | $89.33_{(0.72)}$ | $83.21_{(0.53)}$ |
| *Category II* | | | | | | |
| CGIB | $98.66_{(0.61)}$ | $97.24_{(0.47)}$ | $95.03_{(0.54)}$ | $84.98_{(0.42)}$ | $98.45_{(0.31)}$ | $95.44_{(0.24)}$ |
| CMRL | $98.73_{(0.31)}$ | $96.82_{(0.29)}$ | $94.78_{(0.23)}$ | $84.78_{(0.25)}$ | $98.67_{(0.12)}$ | $95.62_{(0.17)}$ |
| DSN-DDI | $97.87_{(0.14)}$ | $96.29_{(0.13)}$ | $94.37_{(0.16)}$ | $84.30_{(0.08)}$ | $97.31_{(0.10)}$ | $94.34_{(0.08)}$ |
| *Category III* | | | | | | |
| ISE | $98.75_{(0.46)}$ | $97.38_{(0.31)}$ | $94.97_{(0.39)}$ | $85.34_{(0.53)}$ | $98.74_{(0.27)}$ | $95.85_{(0.32)}$ |
| **IGIB-ISE** | $\mathbf{98.97}_{(0.37)}$ | $\mathbf{97.79}_{(0.26)}$ | $\mathbf{95.47}_{(0.21)}$ | $\mathbf{85.93}_{(0.17)}$ | $\mathbf{99.13}_{(0.19)}$ | $\mathbf{96.34}_{(0.12)}$ |

## E.2 SUPPLEMENTARY ABLATION EXPERIMENTS

We conducted additional ablation experiments on three DDI datasets (ZhangDDI, ChChMiner and DeepDDI) and three solvent-solute datasets (FreeSolv, Abraham, and CombiSolv). These experiments

Table 6: Performance of different methods in inductive settings (Bold numbers are the best results).

| Interaction Category | Method | DeepDDI | | ZhangDDI | | ChchMiner | |
|---|---|---|---|---|---|---|---|
| | | AUROC | F1 | AUROC | F1 | AUROC | F1 |
| *Inductive Setting 1* | | | | | | | |
| *Category I* | GoGNN | $71.34_{(1.24)}$ | $67.16_{(1.13)}$ | $63.17_{(1.42)}$ | $45.53_{(1.28)}$ | $69.52_{(1.84)}$ | $69.22_{(1.33)}$ |
| | CIGIN | $72.64_{(1.77)}$ | $69.55_{(1.45)}$ | $68.39_{(1.07)}$ | $44.39_{(1.42)}$ | $77.49_{(1.27)}$ | $75.92_{(1.48)}$ |
| | SSI-DDI | $75.93_{(1.14)}$ | $72.23_{(0.77)}$ | $69.56_{(1.21)}$ | $47.59_{(1.17)}$ | $79.64_{(1.53)}$ | $77.61_{(1.24)}$ |
| | MHCADDI | $68.18_{(0.87)}$ | $67.37_{(1.24)}$ | $62.52_{(0.97)}$ | $44.51_{(1.38)}$ | $70.92_{(1.08)}$ | $75.15_{(0.97)}$ |
| *Category II* | CGIB | $80.80_{(0.53)}$ | $78.47_{(0.57)}$ | $72.80_{(0.43)}$ | $57.29_{(0.58)}$ | $86.41_{(0.93)}$ | $85.13_{(0.43)}$ |
| | CMRL | $84.96_{(0.87)}$ | $77.81_{(0.74)}$ | $74.59_{(1.05)}$ | $56.41_{(0.97)}$ | $87.64_{(0.54)}$ | $86.55_{(0.57)}$ |
| | DSN-DDI | $83.11_{(0.76)}$ | $78.68_{(0.70)}$ | $73.49_{(1.02)}$ | $56.64_{(0.77)}$ | $86.93_{(0.65)}$ | $85.81_{(0.83)}$ |
| *Category III* | ISE | $83.86_{(0.97)}$ | $79.55_{(1.01)}$ | $75.16_{(0.86)}$ | $58.27_{(0.83)}$ | $87.52_{(0.83)}$ | $86.63_{(0.74)}$ |
| | **IGIB-ISE** | $\mathbf{85.01}_{(0.41)}$ | $\mathbf{80.08}_{(0.56)}$ | $\mathbf{75.32}_{(0.32)}$ | $\mathbf{58.96}_{(0.47)}$ | $\mathbf{87.88}_{(0.62)}$ | $\mathbf{87.37}_{(0.54)}$ |
| *Inductive Setting 2* | | | | | | | |
| *Category I* | GoGNN | $64.91_{(3.61)}$ | $68.53_{(3.34)}$ | $54.37_{(2.47)}$ | $34.92_{(3.26)}$ | $67.73_{(3.63)}$ | $72.19_{(4.29)}$ |
| | CIGIN | $68.67_{(3.54)}$ | $69.34_{(4.53)}$ | $57.67_{(2.03)}$ | $33.68_{(4.35)}$ | $65.36_{(2.93)}$ | $71.73_{(3.54)}$ |
| | SSI-DDI | $69.37_{(4.16)}$ | $67.18_{(3.94)}$ | $59.33_{(3.26)}$ | $37.16_{(3.89)}$ | $68.39_{(1.94)}$ | $74.95_{(2.17)}$ |
| | MHCADDI | $63.89_{(3.42)}$ | $63.57_{(5.17)}$ | $56.47_{(2.77)}$ | $33.53_{(3.18)}$ | $63.57_{(4.67)}$ | $64.51_{(4.35)}$ |
| *Category II* | CGIB | $68.78_{(1.67)}$ | $75.72_{(1.93)}$ | $57.24_{(1.97)}$ | $28.83_{(4.53)}$ | $69.82_{(1.52)}$ | $78.46_{(2.03)}$ |
| | CMRL | $73.38_{(1.96)}$ | $73.91_{(2.14)}$ | $60.02_{(2.03)}$ | $40.73_{(3.04)}$ | $69.62_{(1.67)}$ | $75.76_{(1.28)}$ |
| | DSN-DDI | $72.71_{(1.37)}$ | $77.96_{(1.64)}$ | $61.88_{(1.12)}$ | $40.49_{(2.32)}$ | $69.34_{(1.34)}$ | $79.52_{(1.21)}$ |
| *Category III* | ISE | $73.92_{(1.64)}$ | $78.57_{(2.14)}$ | $62.42_{(1.35)}$ | $41.38_{(2.56)}$ | $69.67_{(1.52)}$ | $78.73_{(1.26)}$ |
| | **IGIB-ISE** | $\mathbf{74.51}_{(1.54)}$ | $\mathbf{79.64}_{(1.67)}$ | $\mathbf{63.24}_{(1.72)}$ | $\mathbf{42.03}_{(2.34)}$ | $\mathbf{70.09}_{(1.46)}$ | $\mathbf{80.12}_{(1.48)}$ |

were designed to demonstrate the contribution of each model component across various data scales and task types. We ensured that all experiments followed the same setup (except for the ablated components) and repeated them five times to provide robust results. The results are reported as **Mean (Variance)**.

Table 7: Results on DDI datasets (Evaluation Metric: ACC (%))

| Dataset | $\beta_1 = 0$ | $\beta_2 = 0$ | w/o KL Loss | w/o Contrastive Loss | Baseline |
|---|---|---|---|---|---|
| ZhangDDI | 88.34 (0.41) | 88.39 (0.27) | 88.37 (0.39) | 88.59 (0.24) | **88.84 (0.32)** |
| DeepDDI | 96.27 (0.34) | 96.33 (0.31) | 96.12 (0.28) | 96.41 (0.19) | **96.65 (0.37)** |
| ChChMiner | 94.86 (0.37) | 94.82 (0.11) | 94.93 (0.17) | 95.33 (0.26) | **95.56 (0.28)** |

Table 8: Results on Solvent-Solute datasets (Evaluation Metric: RMSE)

| Dataset | $\beta_1 = 0$ | $\beta_2 = 0$ | w/o KL Loss | w/o Contrastive Loss | Baseline |
|---|---|---|---|---|---|
| FreeSolv | 0.921 (0.058) | 0.886 (0.029) | 0.986 (0.030) | 0.921 (0.033) | **0.713 (0.034)** |
| Abraham | 0.353 (0.002) | 0.419 (0.009) | 0.414 (0.001) | 0.366 (0.001) | **0.343 (0.009)** |
| CombiSolv | 0.411 (0.004) | 0.397 (0.004) | 0.413 (0.001) | 0.411 (0.001) | **0.394 (0.008)** |

As shown in the Table 7 and Table 8, with all components active, our model achieved the best performance across all datasets. When the KL divergence loss ($\mathcal{L}_{com1}$ and $\mathcal{L}_{com2}$), which facilitates the compression of interactive substructures, was removed, the performance declined on all datasets, with FreeSolv and Abraham experiencing the most significant drops. This highlights the critical role of KL divergence loss in guiding the model towards more precise substructure selection, particularly in regression tasks.

On the other hand, removing the contrastive loss ($\mathcal{L}_{con1}$ and $\mathcal{L}_{con2}$) resulted in a marginal performance reduction for most datasets, except for FreeSolv. This phenomenon could be attributed to the robust interaction modeling of our iterative interaction module, which reduces the reliance on contrastive loss. However, for the FreeSolv dataset, where fewer IN were used, the contrastive loss played a more pivotal role, demonstrating the dataset-dependent utility of this component.

Then, we evaluated the impact of setting $\beta_1$ and $\beta_2$ to zero. For DDI datasets, the results indicate that $\beta_1$ and $\beta_2$ have similar contributions, as evidenced by the small margin of performance differences. Nevertheless, for solvent-solute datasets, $\beta_1$ and $\beta_2$ exhibited distinct impacts. This divergence may stem from the inherent asymmetry in solvent-solute interactions, suggesting that the choice of $\beta_1$ and $\beta_2$ requires careful consideration when dealing with asymmetric molecular interactions.

Finally,We explain the objective function under each ablation experiment. The original objective function is:

$$\underset{\mathcal{G}_{s1},\mathcal{G}_{s2}}{\arg\min} - I\left(\mathbf{Y};\mathcal{G}_{s1},\mathcal{G}_{s2}\right) + \beta_1 I\left(\mathcal{G}_1;\mathcal{G}_{s1} \mid \mathcal{G}_{s2}\right) + \beta_2 I\left(\mathcal{G}_2;\mathcal{G}_{s2} \mid \mathcal{G}_{s1}\right)$$

- **When $\beta_1 = 0$:** The objective becomes

$$\underset{\mathcal{G}_{s1},\mathcal{G}_{s2}}{\arg\min} - I\left(\mathbf{Y};\mathcal{G}_{s1},\mathcal{G}_{s2}\right) + \beta_2 I\left(\mathcal{G}_2;\mathcal{G}_{s2} \mid \mathcal{G}_{s1}\right)$$

- **When $\beta_2 = 0$:** The objective becomes

$$\underset{\mathcal{G}_{s1},\mathcal{G}_{s2}}{\arg\min} - I\left(\mathbf{Y};\mathcal{G}_{s1},\mathcal{G}_{s2}\right) + \beta_1 I\left(\mathcal{G}_1;\mathcal{G}_{s1} \mid \mathcal{G}_{s2}\right)$$

- **Without KL loss:** The objective becomes

$$\underset{\mathcal{G}_{s1},\mathcal{G}_{s2}}{\arg\min} - I\left(\mathbf{Y};\mathcal{G}_{s1},\mathcal{G}_{s2}\right) - \beta_1 I\left(\mathcal{G}_{s1};\mathcal{G}_{s2}\right) - \beta_2 I\left(\mathcal{G}_{s2};\mathcal{G}_{s1}\right)$$

- **Without Contrastive loss:** The objective becomes

$$\underset{\mathcal{G}_{s1},\mathcal{G}_{s2}}{\arg\min} - I\left(\mathbf{Y};\mathcal{G}_{s1},\mathcal{G}_{s2}\right) + \beta_1 I\left(\mathcal{G}_{s1};\mathcal{G}_1,\mathcal{G}_{s2}\right) + \beta_2 I\left(\mathcal{G}_{s2};\mathcal{G}_2,\mathcal{G}_{s1}\right)$$

### E.3 PERFORMANCE ON LARGER DATASET

To validate the scalability and effectiveness of our method, we performed experiments on larger datasets. For solvent-solute dataset, we used the CombiSolv-QM Vermeire & Green (2021b) dataset, containing 1 million solvent–solute combinations from 284 solvents and 11,029 solutes. The solute molar masses range from 2.02 g/mol to 1776.89 g/mol. For the DDI task, we extended our evaluation using the Twosides Tatonetti et al. (2012) dataset and combined it with ZhangDDI, ChChDDI, and DeepDDI datasets to create a benchmark of 843,964 unique drug-drug pairs.

Table 9: Performance on Solvent-Solute Dataset

| Model | RMSE |
|---|---|
| CGIB | 0.0976 |
| CRML | 0.0983 |
| IGIB-ISE | **0.0912** |

Table 10: Performance on DDI Dataset

| Model | ACC | F1 | AUROC |
|---|---|---|---|
| CRML | 84.33% | 74.86% | 92.76% |
| CGIB | 84.14% | 74.69% | 92.41% |
| IGIB-ISE | **84.92%** | **75.14%** | **93.89%** |

As shown in Table 9, IGIB-ISE achieves the lowest RMSE of 0.0912, outperforming CGIB and CRML, and highlighting its ability to effectively capture complex solvent-solute interactions. Similarly, Table 10 demonstrates that IGIB-ISE consistently surpasses CGIB and CRML in accuracy (84.92%), F1-score (75.14%), and AUROC (93.89%), underscoring its robustness in identifying intricate drug-drug interactions while maintaining high predictive accuracy. These results underscore IGIB-ISE's superior performance on large datasets, with the lowest RMSE for the solvent-solute task and leading metrics for the DDI task, showcasing its robustness and scalability for real-world applications.

### E.4 OPTIMAL IN FOR DIFFERENT MOLECULAR SCALES

This section investigates the relationship between IN and molecular scale. We combined multiple datasets, including ZhangDDI, ChChDDI, DeepDDI, and Twosides Tatonetti et al. (2012), into a

Table 11: Model performance (ACC) for various IN and AM values.

| IN | AM = 340 | AM = 549 | AM = 638 | AM = 722 | AM = 1934 |
|---|---|---|---|---|---|
| 5 | 78.85% | 72.52% | 69.57% | 69.00% | 84.08% |
| 10 | **79.38%** | 74.93% | 70.98% | 69.45% | 84.77% |
| 20 | 78.98% | **75.47%** | **73.47%** | **70.65%** | 85.02% |
| 30 | 78.25% | 75.14% | 72.13% | 68.90% | **85.24%** |
| 40 | 78.03% | 74.83% | 72.34% | 69.13% | 85.11% |

larger, consolidated dataset. The dataset was categorized into five groups based on the **molar mass** of the molecules, with each group containing 50,000 drug-drug pairs. We then determined the optimal IN for each group by assessing model performance using ACC. The results are summarized in the Table 11, where **AM** represents the average molar mass for each group.

The results demonstrate a clear trend: larger average molar masses (AM) require higher iteration numbers (IN) to achieve optimal performance. For small molecules (AM ≈ 340), an IN value of **10** strikes a balance between computational cost and accuracy. Medium-sized molecules (AM = 549 to 638) benefit most from an IN value of **20**, which effectively improves performance without incurring significant computational overhead. For larger molecules (AM ≈ 722), an IN of **20** remains optimal, as increasing IN to higher values (e.g., 30) can degrade performance. Meanwhile, for very large molecules (AM ≈ 1934), an IN value of **30** fully leverages the model's capability to handle the increased complexity. This flexible approach enables the tailored selection of IN based on molecular scale, ensuring an optimal balance of performance and computational efficiency.

### E.5    SCALABILITY OF IGIB-ISE ACROSS MOLECULAR SIZES

To further assess the scalability of IGIB-ISE, we utilized the dataset from E.4 for performance evaluation. The results are presented in the Table 12.

Table 12: Performance comparison of IGIB-ISE, CGIB, and CMRL across AM.

| Model | AM = 340 | AM = 549 | AM = 638 | AM = 722 | AM = 1934 |
|---|---|---|---|---|---|
| **IGIB-ISE** | **79.38%** | **75.47%** | **73.47%** | **70.65%** | **85.24%** |
| CGIB | 78.14% | 74.31% | 72.59% | 68.91% | 84.62% |
| CMRL | 78.42% | 74.19% | 72.68% | 69.72% | 84.43% |

The results reveal that IGIB-ISE consistently outperforms both CGIB and CMRL across various molecular scales, demonstrating its robustness and scalability. For small molecules (**AM ≈ 340**), IGIB achieves an accuracy of 79.38%, surpassing CGIB and CMRL by 1.24% and 0.96%, respectively. In the medium molecule categories (**AM = 549 to 638**), IGIB attains accuracies of 75.47% and 73.47%, with improvements of approximately 1.2% and 0.8% over the baselines. For large molecules (**AM ≈ 722**), IGIB achieves 70.65%, outperforming CGIB (68.91%) and CMRL (69.72%). Notably, for very large molecules (**AM ≈ 1934**), IGIB reaches a remarkable accuracy of 85.24%, exceeding CGIB (84.62%) and CMRL (84.43%). These findings underscore IGIB's effectiveness as a reliable method for analyzing diverse molecular sizes, making it a robust tool for molecular-scale applications.

### E.6    DYNAMIC ITERATION PROCESS OF THE INTERACTIVE SUBSTRUCTURE EXTRACTION

As illustrated in Figure 5, we present schematic diagrams depicting the substructure interaction processes of Pentazocine and Aminocaproic acid drugs over 40 iterations. Notably, for the Aminocaproic acid drug, it is evident that the amino group is more significant in the early iterations. This observation raises the possibility that, in algorithms selecting substructures based on original molecular graphs, the amino group might be chosen as a core substructure, thereby impacting the final prediction. In contrast, our ISE framework, through iterative steps, selectively identifies the core interaction substructures of Pentazocine. This process determines the insignificance of the amino group, leading

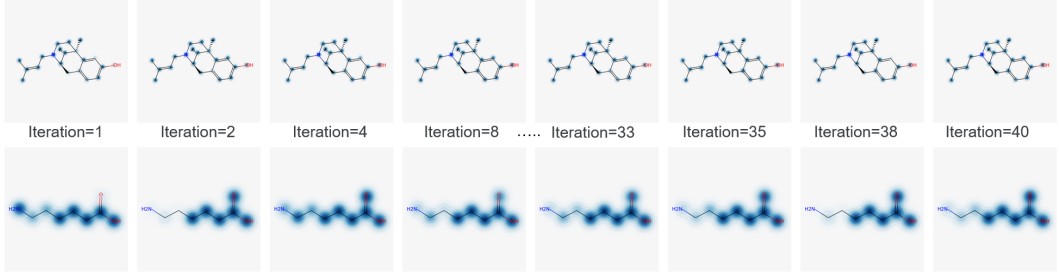

Figure 5: Schematic diagram of the substructure interaction process of Pentazocine (upper) and Aminocaproic acid (lower) drugs.

to its removal in later iterations, and gradually stabilizing the results. Additionally, we also observe that after the EM algorithm finds the optimal substructure, it continues to exhibit slight fluctuations around the converged result.

Table 13: Comparison of time and memory usage during the training phase.

| Model | Metric | ZhangDDI | ChChMiner | DeepDDI | MNSol | FreeSolv | CompSol | Abraham | CombiSolv |
|---|---|---|---|---|---|---|---|---|---|
| CGIB | Memory (GB) | 5.12 | 3.93 | 7.41 | 2.13 | 2.11 | 2.14 | 2.42 | 2.31 |
| | Time (hours) | 1.52 | 0.63 | 3.75 | 0.042 | 0.0034 | 0.081 | 0.164 | 0.153 |
| CMRL | Memory (GB) | **4.03** | **3.45** | **6.12** | **2.12** | **2.12** | **2.13** | **2.45** | **2.33** |
| | Time (hours) | **1.34** | **0.53** | **3.24** | **0.041** | **0.0032** | **0.072** | **0.142** | **0.121** |
| IGIB-ISE | Memory (GB) | 36.2 | 27.3 | 39.4 | 2.12 | 1.83 | 2.25 | 2.64 | 2.42 |
| | Time (hours) | 8.73 | 2.91 | 22.8 | 0.083 | 0.0037 | 0.153 | 0.225 | 0.257 |

Table 14: Comparison of time and memory usage during the inference phase.

| Model | Metric | ZhangDDI | ChChMiner | DeepDDI | MNSol | FreeSolv | CompSol | Abraham | CombiSolv |
|---|---|---|---|---|---|---|---|---|---|
| CGIB | Memory (MB) | 277.5 | 303.2 | 296.7 | 85.34 | 38.41 | 84.97 | 103.4 | 103.8 |
| | Time (s) | 24.8 | 6.81 | 94.7 | 1.69 | 0.94 | 2.62 | 3.64 | 4.76 |
| CMRL | Memory (MB) | **236.2** | **254.2** | **252.3** | **72.37** | **34.21** | **71.43** | **93.87** | **92.34** |
| | Time (s) | 23.5 | **5.89** | 77.3 | **1.49** | **0.88** | 2.31 | **3.43** | **4.37** |
| IGIB-ISE | Memory (MB) | 274.8 | 301.3 | 294.2 | 81.28 | 37.33 | 75.31 | 100.8 | 98.45 |
| | Time (s) | **22.6** | 5.97 | **74.9** | 1.62 | 0.92 | **2.22** | 3.53 | 4.55 |

## F    TIME AND SPACE COMPLEXITY

### F.1    TRAINING PHASE

In this work, we evaluated IGIB-ISE in terms of time and space complexity, dividing the analysis into the training and inference phases across multiple datasets and comparing it with several existing methods. For training phase, as shown in Table 13, while our method effectively reduces redundant information, it incurs higher time and memory overhead compared to baseline models. This is attributed to the extensive use of interaction networks in our iterative substructure selection process, which is integrated with end-to-end optimization to achieve superior performance. This design leads to redundant storage and prolonged gradient computations, contributing to the observed overhead.

- **Time Complexity:** The primary contributor to extended training time is the repeated gradient computations within the interaction network during substructure iterations.
- **Space Complexity:** The redundant storage of interaction network parameters across iterations is the main factor for increased memory usage.

### F.2    INFERENCE PHASE

In real-world applications, inference efficiency is critical. Once trained, the core substructure extractor is directly applied for molecular interaction predictions. For inference phase, as shown in Table 14,

IGIB-ISE demonstrates competitive efficiency. For instance, on large datasets like ZhangDDI, it completes inference in just 22.6 seconds, outperforming CGIB and CMRL. Its memory usage during inference is low and comparable to the baseline methods (e.g., 274.8 MB for ZhangDDI compared to 277.5 MB for CGIB). These results highlight IGIB-ISE's practicality for real-world applications without incurring significant resource overhead.

### F.3 ENGINEERING OPTIMIZATIONS FOR TRAINING EFFICIENCY

In this section, we outline three potential engineering optimizations to improve the efficiency of the **training phase**:

- **Optimization of Computation Graph Storage:** Interaction network parameters, unchanged during iterations, can be globally stored and reused for gradient computation. This reduces redundant storage while maintaining functionality.

- **Core Substructure Initialization:** Reducing iterations by initializing substructures based on prior chemical knowledge (e.g., functional groups) can accelerate convergence and reduce training overhead.

- **Efficient Parameter Fine-Tuning (e.g., LoRA Hu et al. (2021)):** Using low-rank matrices for fine-tuning allows freezing the interaction network and adapting it with minimal computational cost, significantly reducing both memory and time requirements. The pre-trained parameters of the interaction network can be obtained from baseline models such as CGIB.

