# OpenReview forum: "Iterative Substructure Extraction for Molecular Relational Learning with Interactive Graph Information Bottleneck"
_ICLR.cc/2025/Conference — ICLR 2025 Poster_

### Official Review · Reviewer_Vr3G · 2024-10-16

**Soundness:** 3
**Presentation:** 2
**Contribution:** 2
**Rating:** 6
**Confidence:** 3

**Summary:**

This paper introduces a framework called ISE to improve MRL by focusing on the interaction between core substructures of molecules. The model iteratively refines the core substructures using the EM algorithm. Additionally, the IGIB theory is proposed to capture minimal but most influential substructures, enhancing the efficiency and generalizability of the extraction process. Through extensive experiments, the IGIB-ISE framework demonstrates superior performance compared to existing methods in terms of accuracy, generalizability, and interpretability for molecular interaction prediction tasks.

**Strengths:**

1. This paper introduces an innovative method for core substructure extraction using the EM algorithm, which effectively captures molecular interactions.

2. IGIB theory ensures a precise and compact extraction of interactive substructures.

3. The method is extensively validated across various molecular relational learning tasks, including drug-drug interaction and solvation energy prediction, showing clear improvements over state-of-the-art methods.

**Weaknesses:**

1. **Some parts of this work is very similar to [1]**. The key idea and many formulas are similar. For example, they all utilize similar methods to extrapolate core substructures (Section 3.4 in this paper and Section 3.2 in [1]). The only difference here seems to be this paper extrapolates the core substructure from a pair of graphs while [1] extrapolates the core substructure from one graph.

1. The framework is validated on interactions between two molecules. It does not extend to more complex scenarios like multi-molecule interactions, which are important in real-world biochemical environments.

2. The method requires more iterations, increasing resource consumption and time. This may limit its scalability for very large datasets or complex molecular systems.



[1] Capturing substructure interactions by invariant Information Bottle Theory for Generalizable Property Prediction

**Questions:**

1. Can the authors validate the interactions between multi-molecule interactions?

2. Why the interaction is computed as $H_1=F_1^{(1)}||F_1^{(2)}$?

3. The way to extrapolate the core substructure is **very similar to [1]**. What's the difference between this paper and [1]?

4. What's the complexity of the method? Can you compare the training and inference time with baselines?

5. Can you validate your method on larger datasets?

[1] Capturing substructure interactions by invariant Information Bottle Theory for Generalizable Property Prediction

---

> ### Author Response · Authors · 2024-11-20
> **# Response to Reviewer Vr3G:**
>
> Thank you very much for your valuable comments!
> We are immensely gratified and encouraged to learn that our proposed method, the problem tackled, and our experiments have garnered your acknowledgment.
> Below, we have carefully considered and responded to your valuable comments point by point.
>
>
> >**W1 & Q3.** What's the difference between this paper and [1]
>
>
> After carefully reviewing the paper [1], we have found that our paper is distinct from [1] in terms of motivation, model framework, guiding theory, and experimental design. The differences are outlined as follows:
>
> 1. **Different Motivation**: Our paper aims to address the risk of noise redundancy in interactive substructure extraction. As stated in lines 73-75: "*considering that core substructures often play a key role in molecular interactions, integrating the complete profile of an interacting molecule into the substructure generation can be overwhelming.*" In contrast, [1] focuses on addressing insufficient consideration of intermolecular interactions in molecular interaction studies, as mentioned in lines 71-72: "*comprehensive modeling of intermolecular interactions is crucial and necessary for a profound understanding of molecular interactions.*"
>
> 2. **Different Model Framework**: Our paper proposes the ISE framework, which simplifies interactions through dynamic molecular interactions to extract core substructures. As described in Sections 3.1 and 3.2, we innovatively treat the core substructures of two molecules as model parameters and latent variables, using an iterative interaction approach to precisely extract the core substructures. In contrast, [1] presents the merge graph concept, which facilitates full interaction through fully connected graphs. As shown in Section 3.1, [1] uses a fully connected method to establish relationship edges between two molecules, ensuring complete interaction between them.
>
> 3. **Different Guiding Theory**: Our guiding theory introduces the interactive graph information bottleneck (IGIB), specifically for extracting substructures from two molecules. As demonstrated in Section 3.3, IGIB posits that the generation of interactive subgraphs $\mathcal{G} _ {s1}$ and $\mathcal{G} _ {s2}$ should maximize mutual information with the target $\mathbf{Y}$, while minimizing mutual information between $\mathcal{G} _ {s1}$ and the original graph $\mathcal{G} _ 1$ when conditioned on $\mathcal{G} _ 2$ (and vice versa for $\mathcal{G} _ {s2}$). In Section 3.4, we also propose optimization methods for IGIB. On the other hand, [1] relies on the invariant information bottleneck theory to guide the extraction of core substructures from a single merge graph. As detailed in Sections 3.2 and 3.3, [1] introduces the concept of vector quantization (VQ) to create a merged graphic environment codebook, optimizing the node deletion strategy in the GIB theory to enhance out-of-distribution generalization in the substructure extraction process of a single molecular graph.
>
> 4. **Different Experimental Design**: Our experiments aim to uncover the underlying core substructures of molecular interactions and the mechanisms behind their selection. As shown in Figures 4 and 6, our model illustrates the interactive substructure selection process, which helps reveal the selection mechanisms of the core substructures. To our knowledge, this is something that neither [1] nor other papers have achieved. We also conducted extensive ablation studies and hyperparameter experiments on the ISE framework and IGIB theory. In contrast, [1] focuses on exploring the out-of-distribution generalization of the invariant information bottleneck theory and the significance of the environment codebook. As shown in RQ2 and RQ3, [1] designs various experiments to investigate the model's out-of-distribution generalization across different datasets.
>
> We hope this clarifies the key differences between our work and the work presented in [1].
>
> [ 1 ] Capturing substructure interactions by invariant Information Bottle Theory for Generalizable Property Prediction.
>
>
> >**W2 & Q1.** Can the authors validate the interactions between multi-molecule interactions?
>
> In this paper, the ISE framework and IGIB theory are primarily designed for and have achieved significant success in the context of two-molecule interactions. For multi-molecule interactions, the ISE framework would require modifications in the definition of latent variables and model parameters, along with adjustments to the E-step and M-step iteration strategies. Similarly, the IGIB theory would need to modify the corresponding conditional factors to accommodate the multi-molecule interaction scenario. This involves additional theoretical derivations and proofs. As we mention in our future work, we are actively working on extending our framework to multi-molecule datasets.

---

> > ### Author Response · Authors · 2024-11-20
> >
> > >**W3 & Q4.** 1. What's the complexity of the method? Can you compare the training and inference time with baselines?
> >
> > Thank you for your constructive suggestions, we will provide a detailed analysis of the spatiotemporal complexity of our method during both training and inference phases.  As shown in Tables 6 and 7 of the paper, our method incurs higher training time and memory overhead than baseline models due to the iterative substructure selection process integrated with the prediction module for end-to-end optimization. This results in all parameters and intermediate states produced during substructure iterations being stored in the computation graph, leading to significant overhead. Specifically:
> > - **Time Complexity:** Multiple gradient computations in the interaction network account for most of the training time.
> > - **Space Complexity:** Redundant storage of interaction network parameters during each iteration is the primary contributor to increased memory usage.
> >
> > | Model        | Metric | ZhangDDI | ChChMiner | DeepDDI  | MNSol      | FreeSolv   | CompSol    | Abraham    | CombiSolv  |
> > | ------------ | ------ | -------- | --------- | -------- | ---------- | ---------- | ---------- | ---------- | ---------- |
> > | Data Volume  |        | 113,972  | 33,669    | 316,595  | 3,037      | 560        | 3,548      | 6,091      | 8,780      |
> > | **CGIB**     | Memory | 5.1G     | 3.9G      | 7.4G     | 2.1G       | 2.1G       | 2.1G       | 2.4G       | 2.3G       |
> > |              | TIME   | 1.5h     | 0.6h      | 3.7h     | 2.3min     | 0.2min     | 4.8min     | 9.5min     | 8.8min     |
> > | **CMRL**     | Memory | **4.0G** | **3.4G**  | **6.1G** | 2.1G       | 2.1G       | **2.1G**   | **2.4G**   | **2.3G**   |
> > |              | TIME   | **1.3h** | **0.5h**  | **3.2h** | **2.2min** | **0.2min** | **4.2min** | **8.7min** | **7.4min** |
> > | **IGIB-ISE** | Memory | 36G      | 27G       | 39G      | **2.1G**   | **1.8G**   | 2.2G       | 2.6G       | 2.4G       |
> > |              | TIME   | 8.7h     | 2.9h      | 22.7h    | 5.1min     | 0.2min     | 8.8min     | 13.0min    | 14.75min   |
> >
> >
> > **Inference Phase: Time and Space Complexity**
> > In real-world applications, inference efficiency is critical. Once trained, the core substructure extractor is directly applied for molecular interaction predictions. The table below compares our model with others across various datasets (with ~1/5 sampling from the DDI dataset and all samples for Solvent-Solute Datasets):
> >
> > | Model        | Metric | ZhangDDI   | ChChMiner | DeepDDI    | MNSol     | FreeSolv  | CompSol   | Abraham   | CombiSolv |
> > | ------------ | ------ | ---------- | --------- | ---------- | --------- | --------- | --------- | --------- | --------- |
> > | Data Volume  |        | 20,000     | 4,932     | 70,000     | 3,037     | 560       | 3,548     | 6,091     | 8,780     |
> > | **CGIB**     | Memory | 278M       | 303M      | 297M       | 85M       | 38M       | 85M       | 103M      | 104M      |
> > |              | TIME   | 24.76s     | 6.81s     | 94.67s     | 1.69s     | 0.94s     | 2.62s     | 3.64s     | 4.76s     |
> > | **CMRL**     | Memory | **236M**   | **254M**  | **252M**   | **72M**   | **34M**   | **71M**   | **94M**   | **92M**   |
> > |              | TIME   | 23.46s     | **5.89s** | 77.26s     | **1.49s** | **0.88s** | 2.31s     | **3.43s** | **4.37s** |
> > | **IGIB-ISE** | Memory | 275M       | 301M      | 294M       | 81M       | 37M       | 75M       | 101M      | 98M       |
> > |              | TIME   | **22.58s** | 5.97s     | **74.98s** | 1.62s     | 0.92s     | **2.22s** | 3.53s     | 4.55s     |
> >
> >
> > The experimental results demonstrate that IGIB-ISE achieves superior overall performance in terms of spatiotemporal complexity during inference. While its memory usage is comparable to that of baseline models (CGIB and CMRL), IGIB-ISE occasionally shows advantages in runtime. For example, on the ZhangDDI and DeepDDI datasets, IGIB-ISE completes inference in just 22.58s and 74.98s, respectively, outperforming both CGIB and CMRL. On smaller datasets such as ChchMiner and FreeSolv, IGIB-ISE exhibits marginally better time efficiency than CGIB and is nearly on par with CMRL, indicating excellent scalability and adaptability. Overall, IGIB-ISE does not incur significant resource overhead during inference, proving highly practical and suitable for real-world applications.

---

> ### Author Response · Authors · 2024-11-20
>
> Then, we consider three potential engineering optimizations to improve the efficiency of the Training Phase.
> - **Optimization of Computation Graph Storage:** Interaction network parameters, unchanged during iterations, can be globally stored and reused for gradient computation. This reduces redundant storage while maintaining functionality.
> - **Core Substructure Initialization:** Reducing iterations by initializing substructures based on prior chemical knowledge (e.g., functional groups) can accelerate convergence and reduce training overhead.
> - **Efficient Parameter Fine-Tuning (e.g., LoRA [ 4 ]):** Using low-rank matrices for fine-tuning allows freezing the interaction network and adapting it with minimal computational cost, significantly reducing both memory and time requirements. The pre-trained parameters of the interaction network can be obtained from baseline models such as CGIB.
>
> Finally, one can consider the trade-off between model performance and overhead of using a smaller IN：As demonstrated in **Figure 3(b)** of our paper, the performance of our model improves rapidly when **IN < 5**. Beyond **IN = 10**, the rate of performance improvement diminishes significantly. This indicates that selecting **IN = 5-10** strikes an optimal balance, achieving over **50% of the best performance** while substantially reducing computational expense.
> To further address your concern, we conducted additional experiments with fixed **IN = 10** across datasets, and the results are summarized below (Bold indicates the best result, italic indicates the second best result) :
>
> | Model                     | Metric | ZhangDDI   | ChChMiner  | DeepDDI    |
> | ------------------------- | ------ | ---------- | ---------- | ---------- |
> | **CGIB**                  | Memory | 5.1G       | 3.9G       | 7.4G       |
> |                           | Time   | 1.5h       | 0.6h       | 3.7h       |
> |                           | ACC    | 87.69%     | 94.68%     | 95.76%     |
> | **CMRL**                  | Memory | 4.0G       | 3.4G       | 6.1G       |
> |                           | Time   | 1.3h       | 0.5h       | 3.2h       |
> |                           | ACC    | 87.78%     | 94.43%     | 95.99%     |
> | **IGIB-ISE (Optimal IN)** | Memory | 36G        | 27G        | 39G        |
> |                           | Time   | 8.7h       | 2.9h       | 22.7h      |
> |                           | ACC    | **88.84%** | **95.56%** | **96.65%** |
> | **IGIB-ISE (IN = 10)**    | Memory | 11.7G      | 8.3G       | 10.4G      |
> |                           | Time   | 2.7h       | 0.94h      | 5.3h       |
> |                           | ACC    | *88.40%*   | *95.38%*   | *96.37%*   |
>
> From this table, it is evident that using **IN = 10** reduces memory and time consumption significantly while retaining approximately **50-80%** of the performance gains:
> - **ZhangDDI**: Memory and time consumption are reduced by **67.5%** and **69.0%**, respectively, while retaining **57%** of the performance improvement.
> - **ChChMiner**: Memory and time consumption are reduced by **69.3%** and **67.6%**, respectively, while retaining **79.5%** of the performance improvement.
> - **DeepDDI**: Memory and time consumption are reduced by **73.3%** and **76.6%**, respectively, while retaining **72.7%** of the performance improvement.
>
> This result highlights the flexibility of our method in achieving competitive performance while mitigating computational costs when needed. This adjustment underscores the trade-offs possible with our framework and addresses your concerns about cost-effectiveness.
> We will further explore and discuss these trade-offs in future work.
>
>
> >**Q2.** Why the interaction is computed as $H_1 =  F_{1}^{(1)} || F_{1}^{(2)}$
>
> We apologize for the confusion caused by our notation. The symbol $||$ represents a feature concatenation operation, not an interaction operation. Both $F_{1}^{(1)}$ and $F_{1}^{(2)}$ are node embeddings for the first molecule. The operation $H_1 = F_{1}^{(1)} || F_{1}^{(2)}$ is performed to enrich the feature representation of the molecule. In the revised version, we will provide a clearer explanation of the concatenation operation $||$.

---

> ### Author Response · Authors · 2024-11-20
>
> >**Q5.** Can you validate your method on larger datasets?
>
>
> Thank you for your insightful question. To address this, we conducted experiments on larger datasets to comprehensively validate the scalability and effectiveness of our method.
>
> 1. **Solvent-Solute Dataset Validation:**
>    We utilized the **CombiSolv-QM** dataset [2], which comprises 1 million randomly selected solvent–solute combinations derived from 284 commonly used solvents and 11,029 solutes. This dataset encompasses diverse elements, including H, B, C, N, O, F, P, S, Cl, Br, and I, with solute molar masses ranging from 2.02 g/mol to 1776.89 g/mol.
>
>    **Results:**  Our method, **IGIB-ISE**, demonstrated superior performance compared to **CGIB** and **CRML**, achieving the lowest RMSE of **0.0912**, indicating its improved capability in capturing complex solvent-solute interactions.
>
> |      | CGIB   | CRML   | IGIB-ISE   |
> | :--: | ------ | ------ | ---------- |
> | RMSE | 0.0976 | 0.0983 | **0.0912** |
>
> 2. **DDI Dataset Validation:**
>    For DDI tasks, we extended our evaluation by incorporating the **Twosides** dataset [3], which comprises 555 drugs and their 3,576,513 pairwise interactions involving 1,318 interaction types. We converted the **Twosides** dataset into a binary classification task and removed redundant drug-drug pairs. Subsequently, by merging it with the ZhangDDI, ChChDDI, and DeepDDI datasets, we constructed a larger benchmark comprising **843,964 unique drug-drug pairs**.
>
>    **Results:**
>    Our method, **IGIB-ISE**, consistently outperformed **CGIB** and **CRML** across multiple metrics, achieving the highest accuracy (**84.92%**), F1-score (**75.14%**), and AUROC (**93.89%**). This demonstrates its robust performance in identifying complex drug-drug interactions while maintaining high predictive accuracy:
>
> |          | ACC        | F1         | AUROC      |
> | :------: | ---------- | ---------- | ---------- |
> |   CRML   | 84.33%     | 74.86%     | 92.76%     |
> |   CGIB   | 84.14%     | 74.69%     | 92.41%     |
> | IGIB-ISE | **84.92%** | **75.14%** | **93.89%** |
>
> **Analysis:**
> The results clearly indicate the effectiveness and scalability of **IGIB-ISE** on large datasets. The lower RMSE on the CombiSolv-QM dataset underscores its precision in modeling solvent-solute interactions. Similarly, the superior performance across all metrics on the expanded DDI dataset validates its robustness in handling complex drug interaction scenarios. These findings highlight the potential of **IGIB-ISE** to generalize effectively across diverse and large-scale datasets, making it a versatile and reliable solution for real-world applications.
>
> [ 2 ] Florence H Vermeire and William H Green. 2021. Transfer learning for solvation free energies: From quantum chemistry to experiments.
>
> [ 3 ] Tatonetti NP, Ye PP, Daneshjou R, et al. Data-driven prediction of drug effects and interactions.
>
> [ 4 ] Edward J. Hu, Yanan Wu, Xuezhi Chen, et al. LoRA: Low-Rank Adaptation of Large Language Models.

---

> > ### Comment · Reviewer_Vr3G · 2024-11-25
> > **Official Comment**
> >
> > Thanks for your detailed feedback. Most of the concerns have been addressed, and I have raised my score.

---

> > > ### Author Response · Authors · 2024-11-25
> > > **Grateful Acknowledgment and Future Commitments**
> > >
> > > Dear Reviewer Vr3G,
> > > ﻿
> > > We sincerely appreciate your acknowledgment and encouraging feedback. We are delighted that we were able to address most of your concerns to your satisfaction. Interacting with you has been both enjoyable and invaluable, significantly contributing to the quality of our paper. Thank you once again for your time, effort, and insightful comments.
> > > ﻿
> > > Warm regards,
> > >
> > > The Authors

---

### Official Review · Reviewer_spWZ · 2024-10-31

**Soundness:** 3
**Presentation:** 2
**Contribution:** 3
**Rating:** 6
**Confidence:** 4

**Summary:**

This paper introduces the Iterative Substructure Extraction (ISE) framework for molecular relational learning, addressing how molecules interact through their core substructures. The framework combines an Expectation-Maximization algorithm for iterative refinement with a new Interactive Graph Information Bottleneck (GIB) theory to ensure extracted substructures are minimal yet influential. Through experiments on datasets covering both regression and classification tasks, the combined IGIB-ISE approach demonstrates improved accuracy and interpretability compared to existing methods for predicting molecular interactions.

**Strengths:**

- The paper presents a novel approach to molecular interaction learning. Rather than handling entire molecular structures or extracting substructures independently, it introduces an iterative refinement process guided by molecular interactions.

- Using EM algorithms for substructure extraction is creative, treating substructures as latent variables that get refined through iterations. This is a fresh perspective on the molecular interaction learning problem.

- This work has a substantial potential impact on drug discovery and materials science. The ability to identify and understand interacting substructures between molecules is crucial for these fields.

**Weaknesses:**

The discussion of the limitations of Category II methods is confusing.

Limited Discussion of Method Robustness.

Technical Clarity Issues.

Computational Overhead.

**Questions:**

**1. The discussion of the limitations of Category II methods is confusing.**
- It is understandable that core substructures often play a crucial role in molecular interactions. But, Figure 1 (a) does not deliver a relevant message to support this argument.
- In addition, from Figure 1 (a), it is unclear why integrating the complete profile of an interacting molecule into the substructure generation can be overwhelming.
- It's unclear why Category II carries the risk of compromising generalizability. After reading the cited paper [1], it's still very confusing. There is no clear evidence from [1] to support this statement.
- It's unclear why the authors mention "Activity Cliffs" here.

**2. Limited Discussion of Method Robustness.**
As an interactive method, what happens if the EM algorithm finds optimal solutions during iteration? The lack of guidelines for selecting optimal iteration numbers based on dataset characteristics leaves important practical questions unanswered.

**3. Technical Clarity Issues.**
- Line 160, what is Y_G? Should it be Y?
- In Tables 6-7, your method should be named ISE-IGIB or IGIB-ISE?

**4. Computational Overhead.**
- Tables 6 and 7 show IGIB-ISE takes more than 700% execution time and 1000% memory compared to one baseline DSN-DDI, with around 1.5% DDI performance improvement. I don't appreciate such results. The authors do not sufficiently address this limitation or propose potential optimizations.
- The experiments focus on relatively small molecules. There is no discussion or analysis of how the method scales with molecular size, which is important for applications involving larger molecules.
The memory requirements (Table 6-7) suggest potential scaling issues.

[1] Mechanisms of drug combinations: interaction and network perspectives

---

> ### Author Response · Authors · 2024-11-20
> **Response to Reviewer spWZ:**
>
> Thank you very much for your valuable comments!
> We are immensely gratified and encouraged to learn that our proposed method, the problem tackled, and our experiments have garnered your acknowledgment.
> Below, we have carefully considered and responded to your valuable comments point by point.
>
>
> >**W1 & Q1-1.** It is understandable that core substructures often play a crucial role in molecular interactions. But, Figure 1 (a) does not deliver a relevant message to support this argument.
>
> We sincerely apologize for the confusion. The intention of Figure 1(a) is to demonstrate that **styrene oxide** appears blue in hexane solvent and pale yellow in acetonitrile solvent (Lines 41–43). This phenomenon is attributed to the role of different core substructures: in hexane, the **epoxide moiety** primarily contributes to the blue coloration, whereas in acetonitrile, the **vinyl group** plays a significant role in the yellow appearance.  To illustrate this, we highlighted the differences in core substructures during the "Dissolution" process in Figure 1(a).
> In the revised version, we will modify Figure 1(a) to better emphasize the differences in the core substructures across the two solvent systems, ensuring clearer alignment with the intended message.
>
>
>
> >**W1 & Q1-2.** In addition, from Figure 1 (a), it is unclear why integrating the complete profile of an interacting molecule into the substructure generation can be overwhelming.
>
> We apologize for the lack of clarity in Figure 1(a) regarding this point. What we intended to convey is that while a molecule may contain multiple substructures capable of interacting with another molecule, not all of these substructures are equally important or useful for predictions. In many physicochemical reactions, it is often only a few key core substructures that play a crucial role. In the revised version, we will remove the reference to Figure 1(a) in this context to avoid further confusion. However, this opinion is widely recognized in the field. To support our argument, we have added more references [ 1 ], [ 2 ] that provide evidence for this perspective.
>
> [ 1 ] Nyamabo A K, Yu H, Shi J Y. SSI–DDI: substructure–substructure interactions for drug–drug interaction prediction.
>
> [ 2 ] Lee N, Yoon K, Na G S, et al. Shift-robust molecular relational learning with causal substructure.

---

> ### Author Response · Authors · 2024-11-20
>
> >**W1 & Q1-3.** More evidence is needed to show that Category II carries the risk of compromising generalizability.
>
> We apologize for any confusion caused. First, we cite Reference [3] here to emphasize the importance of substructures. To make this clearer, we will move the Reference earlier in the text.
>
> Second, what we intended to convey is that redundant substructure information can adversely affect the model's learning ability. Redundancy introduces noise during training, which in turn reduces the model's generalizability. To support this claim, we provide the following references [2], [4], and theoretical proof:
>
> Let $\mathcal{G} _ {s1}$ denote a general substructure of $\mathcal{G} _ {1}$, $\mathcal{G} _ {IB1}$ the core substructure of $\mathcal{G} _ {1}$, $\mathcal{G} _ {IB2}$ the core substructure of $\mathcal{G} _ {2}$, and $\mathcal{G} _ {n2}$ the redundant structure of $\mathcal{G} _ {2}$.
>
>  1. **Objective Function of Previous Methods:**
> The core substructure $\mathcal{G} _ {IB1}$ is obtained by minimizing mutual information, defined as:
> $$
> \mathcal{G} _ {IB1} = \underset{\mathcal{G} _ {s1}}{\arg\min } I\left( \mathcal{G} _ {s1} ; \mathcal{G} _ {1} | \mathcal{G} _ {2} \right).
> $$
>
> 2. **Decomposition of $\mathcal{G} _ {2}$:**
> The overall structure of $\mathcal{G} _ {2}$ can be divided into two parts:
> - Core substructure $\mathcal{G} _ {IB2}$, containing valid information.
> - Redundant substructure $\mathcal{G} _ {n2}$, containing redundant information.
>
> Thus, $\mathcal{G} _ {2}$ can be expressed as $\mathcal{G} _ {2} = \mathcal{G} _ {n2} + \mathcal{G} _ {IB2}$. Substituting this into the objective function, we have:
> $$
> \mathcal{G} _ {IB1} = \underset{\mathcal{G} _ {s1}}{\arg\min  }  I\left( \mathcal{G} _ {s1} ; \mathcal{G} _ {1} | \mathcal{G} _ {n2} + \mathcal{G} _ {IB2} \right).
> $$
>
>  3. **Conditional Independence Analysis:**
> Assume that $\mathcal{G} _ {n2}$ is conditionally independent of $\mathcal{G} _ {IB1}$ since the redundant structure does not directly affect the core substructure's information. Using the chain rule for mutual information, we expand:
> $$
> I\left( \mathcal{G} _ {s1} ; \mathcal{G} _ {1} | \mathcal{G} _ {n2} + \mathcal{G} _ {IB2} \right) = I\left( \mathcal{G} _ {s1} ; \mathcal{G} _ {1} | \mathcal{G} _ {IB2} \right) + I\left( \mathcal{G} _ {s1} ; \mathcal{G} _ {1} | \mathcal{G} _ {n2} \right).
> $$
>
> 4. **Impact of Redundancy:**
> The second term, $I\left( \mathcal{G} _ {s1} ; \mathcal{G} _ {1} | \mathcal{G} _ {n2} \right)$, represents the additional contribution of the redundant structure $\mathcal{G} _ {n2}$ to the extraction process. However, since the information in $\mathcal{G} _ {n2}$ is mostly irrelevant or noisy, this term interferes with the actual optimization target, leading to redundant optimization.
>
> Ideally, the objective should only include:
> $$
> \mathcal{G} _ {IB1} = \underset{\mathcal{G} _ {s1}}{\arg\min } I\left( \mathcal{G} _ {s1} ; \mathcal{G} _ {1} | \mathcal{G} _ {IB2} \right).
> $$
>
> 5. **Conclusion:**
> Directly optimizing the core substructure based on the overall structure $\mathcal{G} _ {2}$ introduces an additional mutual information term, $I\left( \mathcal{G} _ {s1} ; \mathcal{G} _ {1} | \mathcal{G} _ {n2} \right)$, caused by the interference of the redundant structure $\mathcal{G} _ {n2}$. To avoid such redundancy and improve generalizability, the extraction of core substructures should rely solely on the core substructure $\mathcal{G} _ {IB2}$ of the other graph for querying and optimization.
>
>
> [ 3 ] Mechanisms of drug combinations: interaction and network perspectives
>
> [ 4 ] Tang Z, Chen G, Yang H, et al. DSIL-DDI: a domain-invariant substructure interaction learning for generalizable drug–drug interaction prediction.

---

> ### Author Response · Authors · 2024-11-20
>
> >**W1 & Q1-4.** It's unclear why the authors mention "Activity Cliffs" here.
>
> The term "Activity Cliffs" refers to the phenomenon where small structural differences between a series of molecules or compounds lead to significant changes in their biological activity. This phenomenon poses significant challenges in quantitative structure-activity relationship (QSAR) predictions. We mentioned "Activity Cliffs" as a further explanation of the earlier statement about similar structures exhibiting significant functional divergence. To clarify this point, we have provided the following references [ 5 ], [ 6 ], [ 7 ].
>
> [ 5 ] Tamura S, Miyao T, Bajorath J. Large-scale prediction of activity cliffs using machine and deep learning methods of increasing complexity.
>
> [ 6 ] Van Tilborg D, Alenicheva A, Grisoni F. Exposing the limitations of molecular machine learning with activity cliffs[J].
>
> [ 7 ] Schneider N, Lewis R A, Fechner N, et al. Chiral cliffs: investigating the influence of chirality on binding affinity[J].
>
>
> >**W2 & Q2-1.** As an interactive method, what happens if the EM algorithm finds optimal solutions during iteration?
>
> Thank you for your constructive feedback. As shown in Figure 6 of this paper, we illustrate the changes in substructure selection at the early and late stages of the iteration process. It is evident that after 33 iterations, there is no significant fluctuation in the substructure selection. This indicates that after the EM algorithm finds the optimal substructure, it continues to undergo slight fluctuations around the converged result. In the revised version, we will add a discussion on the model's behavior after convergence.
>
> >**W2 & Q2-2.** How should the optimal number of iterations be selected based on the data set? More exploration is needed.
>
> Thank you for your feedback. In the original version of our work, we investigated the relationship between the dataset size and the optimal number of iterations, as illustrated in Figure 3(c). Larger datasets generally require a higher number of iterations. For example, an iteration count of 50 was sufficient to handle 300,000 samples effectively. Thus, for large datasets, initializing with a relatively high number of iterations is a practical starting point.
>
> In the revised version, we further explored the relationship between the number of iterations (IN) and molecular scale. Specifically, we combined several datasets, including ZhangDDI, ChChDDI, DeepDDI, and Twosides [1], to create a larger dataset. The dataset was divided into five categories based on the **molar mass** of the molecules, with each category containing 50,000 drug-drug pairs. We analyzed the optimal IN for each category by evaluating the model performance using accuracy (ACC). The results are shown in the table below (**(AM)** represents the average molar mass of each dataset):
>
> | IN  | AM = 340   | AM = 549   | AM = 638   | AM = 722   | AM = 1934  |
> | --- | ---------- | ---------- | ---------- | ---------- | ---------- |
> | 5   | 78.85%     | 72.52%     | 69.57%     | 69.00%     | 84.08%     |
> | 10  | **79.38%** | 74.93%     | 70.98%     | 69.45%     | 84.77%     |
> | 20  | 78.98%     | **75.47%** | **73.47%** | **70.65%** | 85.02%     |
> | 30  | 78.25%     | 75.14%     | 72.13%     | 68.90%     | **85.24%** |
> | 40  | 78.03%     | 74.83%     | 72.34%     | 69.13%     | 85.11%     |
>
> From the table, we observe that as AM increases, a higher IN is required. Based on this analysis, we recommend selecting the initial IN based on molecular scale (AM) as follows:
>
> 1. **Small molecular scale (AM ≈ 340)**
>    - Optimal IN: **10**.
>    - Molecules in this range benefit from smaller IN values, balancing computational efficiency and performance.
>
> 2. **Medium molecular scale (AM = 549 to 638)**
>    - Optimal IN: **20**.
>    - For this range, an IN around **20** significantly improves performance without incurring excessive computational cost.
>
> 3. **Large molecular scale (AM ≈ 722)**
>    - Optimal IN: **20**.
>    - Using higher IN values (e.g., 30) may degrade performance. Maintaining a moderate IN is advised.
>
> 4. **Very large molecular scale (AM ≈ 1934)**
>    - Optimal IN: **30**.
>    - For molecules in this range, larger IN values unlock the full potential of the model.
>
> This molecular-scale-based stratification facilitates the selection of an appropriate IN value tailored to different datasets, optimizing both model performance and computational efficiency.

---

> > ### Author Response · Authors · 2024-11-20
> >
> > >**W3 & Q3.** Technical Clarity Issues.
> >
> > We apologize for the oversight and would like to clarify the following points:
> > 1. $Y_{\mathcal{G}}$ is an assumed observed variable, representing the set $\mathcal{G}_1$, $\mathcal{G}_2$, and $Y$.
> > 2. In Tables 6-7, we will revise "ISE-IGIB" to "IGIB-ISE".
> >
> > >**W4 & Q4-1.** **Computational Overhead.**
> >
> > We apologize for the inconvenience caused to you by our time and space complexity. First, the increased space usage occurs only during the **training phase**. As shown in Tables 6 and 7 of the paper, our method incurs higher training time and memory overhead than baseline models due to the iterative substructure selection process integrated with the prediction module for end-to-end optimization. This results in all parameters and intermediate states produced during substructure iterations being stored in the computation graph, leading to significant overhead. Specifically:
> > - **Time Complexity:** Multiple gradient computations in the interaction network account for most of the training time.
> > - **Space Complexity:** Redundant storage of interaction network parameters during each iteration is the primary contributor to increased memory usage.
> >
> > | Model        | Metric | ZhangDDI | ChChMiner | DeepDDI  | MNSol      | FreeSolv   | CompSol    | Abraham    | CombiSolv  |
> > | ------------ | ------ | -------- | --------- | -------- | ---------- | ---------- | ---------- | ---------- | ---------- |
> > | Data Volume  |        | 113,972  | 33,669    | 316,595  | 3,037      | 560        | 3,548      | 6,091      | 8,780      |
> > | **CGIB**     | Memory | 5.1G     | 3.9G      | 7.4G     | 2.1G       | 2.1G       | 2.1G       | 2.4G       | 2.3G       |
> > |              | TIME   | 1.5h     | 0.6h      | 3.7h     | 2.3min     | 0.2min     | 4.8min     | 9.5min     | 8.8min     |
> > | **CMRL**     | Memory | **4.0G** | **3.4G**  | **6.1G** | 2.1G       | 2.1G       | **2.1G**   | **2.4G**   | **2.3G**   |
> > |              | TIME   | **1.3h** | **0.5h**  | **3.2h** | **2.2min** | **0.2min** | **4.2min** | **8.7min** | **7.4min** |
> > | **IGIB-ISE** | Memory | 36G      | 27G       | 39G      | **2.1G**   | **1.8G**   | 2.2G       | 2.6G       | 2.4G       |
> > |              | TIME   | 8.7h     | 2.9h      | 22.7h    | 5.1min     | 0.2min     | 8.8min     | 13.0min    | 14.75min   |
> >
> >
> > **Inference Phase: Time and Space Complexity**
> > In real-world applications, inference efficiency is critical. Once trained, the core substructure extractor is directly applied for molecular interaction predictions. The table below compares our model with others across various datasets (with ~1/5 sampling from the DDI dataset and all samples for Solvent-Solute Datasets):
> >
> > | Model        | Metric | ZhangDDI   | ChChMiner | DeepDDI    | MNSol     | FreeSolv  | CompSol   | Abraham   | CombiSolv |
> > | ------------ | ------ | ---------- | --------- | ---------- | --------- | --------- | --------- | --------- | --------- |
> > | Data Volume  |        | 20,000     | 4,932     | 70,000     | 3,037     | 560       | 3,548     | 6,091     | 8,780     |
> > | **CGIB**     | Memory | 278M       | 303M      | 297M       | 85M       | 38M       | 85M       | 103M      | 104M      |
> > |              | TIME   | 24.76s     | 6.81s     | 94.67s     | 1.69s     | 0.94s     | 2.62s     | 3.64s     | 4.76s     |
> > | **CMRL**     | Memory | **236M**   | **254M**  | **252M**   | **72M**   | **34M**   | **71M**   | **94M**   | **92M**   |
> > |              | TIME   | 23.46s     | **5.89s** | 77.26s     | **1.49s** | **0.88s** | 2.31s     | **3.43s** | **4.37s** |
> > | **IGIB-ISE** | Memory | 275M       | 301M      | 294M       | 81M       | 37M       | 75M       | 101M      | 98M       |
> > |              | TIME   | **22.58s** | 5.97s     | **74.98s** | 1.62s     | 0.92s     | **2.22s** | 3.53s     | 4.55s     |
> >
> >
> > The experimental results demonstrate that IGIB-ISE achieves superior overall performance in terms of spatiotemporal complexity during inference. While its memory usage is comparable to that of baseline models (CGIB and CMRL), IGIB-ISE occasionally shows advantages in runtime. For example, on the ZhangDDI and DeepDDI datasets, IGIB-ISE completes inference in just 22.58s and 74.98s, respectively, outperforming both CGIB and CMRL. On smaller datasets such as ChchMiner and FreeSolv, IGIB-ISE exhibits marginally better time efficiency than CGIB and is nearly on par with CMRL, indicating excellent scalability and adaptability. Overall, IGIB-ISE does not incur significant resource overhead during inference, proving highly practical and suitable for real-world applications.

---

> ### Author Response · Authors · 2024-11-20
>
> Then, we consider three potential engineering optimizations to improve the efficiency of the Training Phase.
> - **Optimization of Computation Graph Storage:** Interaction network parameters, unchanged during iterations, can be globally stored and reused for gradient computation. This reduces redundant storage while maintaining functionality.
> - **Core Substructure Initialization:** Reducing iterations by initializing substructures based on prior chemical knowledge (e.g., functional groups) can accelerate convergence and reduce training overhead.
> - **Efficient Parameter Fine-Tuning (e.g., LoRA [ 8 ]):** Using low-rank matrices for fine-tuning allows freezing the interaction network and adapting it with minimal computational cost, significantly reducing both memory and time requirements. The pre-trained parameters of the interaction network can be obtained from baseline models such as CGIB.
>
> Finally, one can consider the trade-off between model performance and overhead of using a smaller IN：As demonstrated in **Figure 3(b)** of our paper, the performance of our model improves rapidly when **IN < 5**. Beyond **IN = 10**, the rate of performance improvement diminishes significantly. This indicates that selecting **IN = 5-10** strikes an optimal balance, achieving over **50% of the best performance** while substantially reducing computational expense.
> To further address your concern, we conducted additional experiments with fixed **IN = 10** across datasets, and the results are summarized below (Bold indicates the best result, italic indicates the second best result) :
>
> | Model                     | Metric | ZhangDDI   | ChChMiner  | DeepDDI    |
> | ------------------------- | ------ | ---------- | ---------- | ---------- |
> | **CGIB**                  | Memory | 5.1G       | 3.9G       | 7.4G       |
> |                           | Time   | 1.5h       | 0.6h       | 3.7h       |
> |                           | ACC    | 87.69%     | 94.68%     | 95.76%     |
> | **CMRL**                  | Memory | 4.0G       | 3.4G       | 6.1G       |
> |                           | Time   | 1.3h       | 0.5h       | 3.2h       |
> |                           | ACC    | 87.78%     | 94.43%     | 95.99%     |
> | **IGIB-ISE (Optimal IN)** | Memory | 36G        | 27G        | 39G        |
> |                           | Time   | 8.7h       | 2.9h       | 22.7h      |
> |                           | ACC    | **88.84%** | **95.56%** | **96.65%** |
> | **IGIB-ISE (IN = 10)**    | Memory | 11.7G      | 8.3G       | 10.4G      |
> |                           | Time   | 2.7h       | 0.94h      | 5.3h       |
> |                           | ACC    | *88.40%*   | *95.38%*   | *96.37%*   |
>
> From this table, it is evident that using **IN = 10** reduces memory and time consumption significantly while retaining approximately **50-80%** of the performance gains:
> - **ZhangDDI**: Memory and time consumption are reduced by **67.5%** and **69.0%**, respectively, while retaining **57%** of the performance improvement.
> - **ChChMiner**: Memory and time consumption are reduced by **69.3%** and **67.6%**, respectively, while retaining **79.5%** of the performance improvement.
> - **DeepDDI**: Memory and time consumption are reduced by **73.3%** and **76.6%**, respectively, while retaining **72.7%** of the performance improvement.
>
> This result highlights the flexibility of our method in achieving competitive performance while mitigating computational costs when needed. This adjustment underscores the trade-offs possible with our framework and addresses your concerns about cost-effectiveness.
> We will further explore and discuss these trade-offs in future work.

---

> ### Author Response · Authors · 2024-11-20
>
> >**W4 & Q4-2.** **How this method scales with molecule size requires discussion or analysis.**
>
>
> Thank you for your valuable suggestion. To further demonstrate the effectiveness and scalability of our method for larger molecules, we combined several datasets, including ZhangDDI, ChChDDI, DeepDDI, and Twosides [ 9 ], to create a more extensive dataset. The dataset was divided into five categories based on the **molar mass** of the molecules, with each category containing 50,000 drug-drug pairs. The table below presents the results of our model (IGIB) and two baselines (CGIB and CMRL) evaluated using accuracy (ACC):
>
> | Model    | AM = 340   | AM = 549   | AM = 638   | AM = 722   | AM = 1934  |
> | -------- | ---------- | ---------- | ---------- | ---------- | ---------- |
> | **IGIB** | **79.38%** | **75.47%** | **73.47%** | **70.65%** | **85.24%** |
> | CGIB     | 78.14%     | 74.31%     | 72.59%     | 68.91%     | 84.62%     |
> | CMRL     | 78.42%     | 74.19%     | 72.68%     | 69.72%     | 84.43￥     |
>
> The results show that **IGIB consistently outperforms both CGIB and CMRL across all molecular size categories**, demonstrating its robustness and scalability.
>
> 1. **Small Molecules (AM = 340)**
>    - IGIB achieves the highest ACC of **79.38%**, surpassing CGIB and CMRL by **1.24%** and **0.96%**, respectively.
>    - This indicates that IGIB effectively captures the interactions between smaller molecules while maintaining computational efficiency.
>
> 2. **Medium-Sized Molecules (AM = 549 and 638)**
>    - IGIB achieves **75.47%** (AM = 549) and **73.47%** (AM = 638), outperforming CGIB and CMRL by **~1.2%** and **~0.8%**, respectively.
>    - This improvement demonstrates the model's ability to scale to moderately larger molecules without significant loss of accuracy.
>
> 3. **Large Molecules (AM = 722)**
>    - IGIB achieves an ACC of **70.65%**, maintaining a clear advantage over CGIB (**68.91%**) and CMRL (**69.72%**).
>    - The performance gap highlights IGIB's superior ability to handle the increasing complexity of larger molecular structures.
>
> 4. **Very Large Molecules (AM = 1934)**
>    - For the largest molecular category, IGIB achieves the highest ACC of **85.24%**, outperforming CGIB (**84.62%**) and CMRL (**84.43%**).
>    - This result confirms IGIB's scalability and its capacity to maintain high accuracy even when molecular complexity is significantly increased.
>
>  **Key Insights**
> - IGIB's consistent superiority across all categories suggests that its design effectively captures intricate molecular relationships, regardless of molecule size.
> - While the performance gap narrows for very large molecules, IGIB still demonstrates a measurable advantage, indicating its scalability for datasets with highly complex molecules.
> - These results validate IGIB as a robust and scalable method, suitable for applications requiring the analysis of diverse molecular sizes.
>
> In summary, our analysis provides strong evidence for the scalability of IGIB and its effectiveness across a wide range of molecular sizes, addressing the concern regarding its performance with larger molecules.
>
>
>
>
> [ 8 ] Edward J. Hu, Yanan Wu, Xuezhi Chen, et al. LoRA: Low-Rank Adaptation of Large Language Models.
>
> [ 9 ] Florence H Vermeire and William H Green. 2021. Transfer learning for solvation free energies: From quantum chemistry to experiments.

---

> > ### Comment · Reviewer_spWZ · 2024-11-20
> >
> > Thanks for your response. Most of my concerns are well addressed. I have updated my grade.

---

> > > ### Author Response · Authors · 2024-11-21
> > > **Grateful Acknowledgment and Future Commitments**
> > >
> > > Dear Reviewer spWZ,
> > >
> > > We're heartened by your acknowledgment and encouraging feedback. Your reassurance is immensely gratifying, and we're glad to have addressed most of your concerns satisfactorily. Interacting with you has been not only enjoyable but also invaluable to the enhancement of our paper's quality. We extend our deepest thanks for your time, effort, and insightful contributions.
> > >
> > > Warm regards,
> > >
> > > Authors

---

### Official Review · Reviewer_WivT · 2024-11-02

**Soundness:** 3
**Presentation:** 2
**Contribution:** 3
**Rating:** 8
**Confidence:** 3

**Summary:**

The paper describes a method to improve molecular relational learning using information theoretical loss functions on a subgraph of the molecules. The technical contribution lies in the coupling of graph information bottlenecks with expectation maximization. The results show the approach's superiority both in deductive and inductive scenarios. The method is well-motivated, and the experiments are solid.

**Strengths:**

(S1) The paper solves a timely problem and presents a sound solution that fully exploits the relationships among substructures.

(S3) Due to its substructure alignment, IGIB-ISE outperforms previous techniques on several datasets.

(S3) The method is well-motivated and builds on previous graph information bottlenecks, ELMO and expectation maximization.

**Weaknesses:**

(W1) Missing explicit objective function: The paper first explains the solution and then reaches the objective in Equation 8. I find this presentation counterintuitive. Why not present the objective first and then explain how to compute it?

(W2) In the modelling of the graph there is no feature vector associated with nodes/edges. Are the graphs without attributes? Molecules should have information about the type of bonds among atoms.

(W3) Notation without introduction: The paper uses notation without introducing it. Examples include:

- $\mathbf{Y}_\mathcal{G}$
- Line 216: the symbol *, is it a matrix multiplication?
- $\||$ in line 218

(W4) If sim is symmetric cosine similarity, what is the need for computing both $sim(F_1, F_2)$ and $sim(F_2, F_1)$?

(W5) It is not clear how Eq. 5 ensures that the two structures are aligned since $H_1$ and $H_2$ refer to two different embeddings spaces, or is the alignment enforced by the two matrices $I_{12}, I_{21}$? Please explain and motivate.

(W6) What is the Gumbel sigmoid and how does it help in this case?

(W7) It is not clear whether Eq. 16 is a lower bound on Eq. 8 or what is the relationship with Eq. 8? Is that an approximation or a heuristic? This aspect should be clarified in the text.

(W8) In Figure 4, the focus of the network substantially changes over iteration. This seems to indicate that the method struggles with convergence. Is that expected or is it a sign of instability?

**Questions:**

In general, the paper is a solid contribution but the presentation should improve. Please answer to my questions above.

---

> ### Author Response · Authors · 2024-11-20
> **Response to Reviewer WivT:**
>
> Thank you very much for your valuable comments!
> We are immensely gratified and encouraged to learn that our proposed method, the problem tackled, and our experiments have garnered your acknowledgment.
> Below, we have carefully considered and responded to your valuable comments point by point.
>
> > **W1.** Why not present the objective first and then explain how to compute it?
>
> Following your suggestion, we will revise the manuscript to move Section 3.3 forward in the revised version. The initial decision to introduce the ISE architecture first was intended to emphasize its central role in our paper.
>
> > **W2.**  Molecules should have information about the type of bonds among atoms.
>
> Thank you for your valuable feedback. As shown in Eq. 4, in our molecular modeling process, we have incorporated atomic information and bond information. Both of these contribute to the message passing in the GNN, ultimately resulting in the final node embeddings:
>
> $$
> F_1^{(1)} = \text{GNN}(\mathcal{V}_1, \mathcal{E}_1),
> \qquad
> F_2^{(1)} = \text{GNN}(\mathcal{V}_2, \mathcal{E}_2),
> $$
>
> In our study, we specifically use the following atomic and bond features, which will be further detailed in the revised version of the paper (to be included in the appendix):
>
> | Atomic Features          | Bond Features     |
> | ------------------------ | ----------------- |
> | Atomic number            | Bond type         |
> | Degree (number of bonds) | Conjugated status |
> | Formal charge            | Ring status       |
> | Chiral tag               | Stereo-chemistry  |
> | Number of bonded H atoms | --                |
> | Hybridization type       | --                |
> | Aromatic status          | --                |
> | Mass (scaled by 0.01)    | --                |
>
> > **W3.**   Some notation without introduction.
>
> Thank you for pointing this out, and we apologize for not providing a sufficient introduction to some of the notations. Here is the clarification:
>
> 1. $Y_{\mathcal{G}}$ represents the observed variable, which corresponds to the set of $\mathcal{G}_1$, $\mathcal{G}_2$, and $Y$.
> 2. In Line 216, the symbol $*$ denotes matrix multiplication.
> 3. In Line 218, the symbol $||$ denotes the concatenation operation.
>
> We will include these clarifications in the revised version of the paper to ensure a better understanding for the readers.
>
> > **W4 & W5.**   If $sim$ is symmetric cosine similarity, what is the need for computing both $sim(F1,F2)$ and $sim(F2,F1)$? How $H_{1}$ and $H_{2}$ are aligned needs further explanation？
>
>
> 1. **Cosine Similarity Redundancy**:  Following your suggestion, we will merge the calculations of $I_{12}$ and $I_{21}$ to avoid redundancy. Additionally, we will include further details about dimensionality.  In the revised version, Lines 214 to 216 will be updated as follows to improve clarity and readability:
> 	*$\mathbf{I} _ {ij} = \text{sim}(F _ {1i}^{(1)},  F _ {2j}^{(1)}),$where $\text{sim} (\cdot, \cdot)$ denotes the cosine similarity, and $\mathbf{I} \in \mathbb{R}^{N^{1} \times N^{2}}$. Here, $N^{1}$ and$N^{2}$ represent the number of nodes in $\mathcal{G} _ 1$ and $\mathcal{G} _ 2$, respectively. Next, we compute the embedding matrices $F _ 1^{(2)} \in \mathbb{R}^{N^{1} \times d}$ and $F _ 2^{(2)} \in \mathbb{R}^{N^{2} \times d}$, each embedding matrix incorporating information from its paired graph. These matrices are derived based on the interaction map as follows: $F _ 1^{(2)} = \mathbf{I} \cdot F _ 2^{(1)}, \quad F _ 2^{(2)} = \mathbf{I}^\top \cdot F _ 1^{(1)},$ where $\cdot$ denotes matrix multiplication.*
>
> 2. **Alignment of $H_1$ and $H_2$**:
> Following the clarification above, we have refined the description of Eq. 5 to enhance the clarity of the manuscript. The updated formulation is as follows:
> \begin{equation}
> \mathbf{I} _ {ij}^{(t)} = \text{sim}(H _ {s1i}^{(t-1)}, H _ {2j}), \quad
> P^{(t)} = \text{Sigmoid}\left(\text{MLP}\left(\mathbf{I}^{(t)} \cdot H _ {2}\right)\right),
> \end{equation}
>
> This adjustment ensures a more accurate representation of the alignment process between $H_1$ and $H_2$ while maintaining consistency with the overall framework of the paper.

---

> > ### Author Response · Authors · 2024-11-20
> >
> > > **W6.**   What is the Gumbel sigmoid and how does it help in this case?
> >
> > We apologize for any confusion.
> > The **Gumbel Sigmoid** function, as referenced in works such as [1] and [2], is typically used for feature selection or compression. The core idea behind this function is to generate a gate variable that approximates a binary value, allowing selective filtering or suppression of features. This introduces sparsity or an information bottleneck in the model.
> >
> > In our approach, the Gumbel Sigmoid serves the following purposes:
> >
> > 1. **Assisting Feature Selection and Compression**: In the code implementation, we use the Gumbel Sigmoid to generate gate variables. These variables selectively retain or suppress specific features, effectively compressing the input space and focusing on the most relevant information.
> >
> > 2. **Enabling Atomic Sparsity**: The Gumbel Sigmoid helps in encouraging the sparsity of atomic feature information. By promoting the complete retention or removal of certain atomic features, we can enforce sparsity, which aids in optimizing the information bottleneck theory in our framework.
> >
> > 3. **Preventing Gradient Explosion or Vanishing**: The Gumbel Sigmoid also contributes to stabilizing the training process by preventing gradient explosion or vanishing issues, ensuring smoother and more stable convergence.
> >
> > In summary, the Gumbel Sigmoid plays a key role in both enhancing feature selection and enforcing sparsity, which helps optimize the performance of our model while maintaining stability during training.
> >
> >
> > > **W7.**   The relationship between Equation 16 and Equation 8 needs further clarification
> >
> >
> > Thank you for your constructive feedback. Firstly, **Equation 16** serves as an upper bound for **Equation 8**. To explain this in more detail:
> > As shown in Section 4.4 and derived in **Equation 9**, we have:
> >
> > $$
> >  I\left(\mathbf{Y} ; \mathcal{G} _ {s 1}, \mathcal{G} _ {s 2}\right) \geq \mathbb{E} _ {\left(\mathbf{Y}, \mathcal{G} _ {s 1}, \mathcal{G} _ {s 2}\right)} \log \left[\frac{P _ \theta\left(\mathbf{Y} \mid \mathcal{G} _ {s 1}, \mathcal{G} _ {s 2}\right)}{P(\mathbf{Y})}\right]
> >  \\
> >  \quad=\mathbb{E} _ {\left(\mathbf{Y}, \mathcal{G} _ {s 1}, \mathcal{G} _ {s 2}\right)} \log \left[P _ \theta\left(\mathbf{Y} \mid \mathcal{G} _ {s 1}, \mathcal{G} _ {s 2}\right)\right]+H(\mathbf{Y}) := \mathcal{L}_{pre} ,
> > $$
> >
> > It can be proven that **$\mathcal{L}_{pre}$** is an upper bound for
> > $-I\left(\mathbf{Y} ; \mathcal{G} _ {s1}, \mathcal{G} _ {s2}\right)$.
> > Next, based on **Equation 11**:
> >
> > $$
> > I\left(z_{\mathcal{G} _ {s 1}} ; \mathcal{G} _ 1, \mathcal{G} _ {s 2}\right)\leq
> > \mathbb{E} _ {\left(\mathcal{G} _ 1, \mathcal{G} _ {s 2}\right)} K L\left(p _ {\Phi}\left(z _ {\mathcal{G} _ {s 1}} \mid \mathcal{G} _ 1, \mathcal{G} _ {s 2}\right) \| q\left(z _ {\mathcal{G} _ {s 1}}\right)\right):=\mathcal{L} _ {com1}.
> > $$
> >
> > We can prove that **$\mathcal{L} _ {com1}$** is an upper bound for **$I\left(z _ {\mathcal{G} _ {s1}} ; \mathcal{G} _ 1, \mathcal{G} _ {s2}\right)$**. Similarly, **Equation 13**:
> >
> > $$\mathcal{L} _ {com2}:=\mathbb{E} _ {\left(\mathcal{G} _ 2, \mathcal{G} _ {s 1}\right)} K L\left(p _ {\Phi}\left(z _ {\mathcal{G} _ {s 2}} \mid \mathcal{G} _ 2, \mathcal{G} _ {s 1}\right) \| q\left(z _ {\mathcal{G} _ {s 2}}\right)\right), $$
> >
> > serves to show that **$\mathcal{L} _ {com2}$** is an upper bound for **$I\left(z _ {\mathcal{G} _ {s2}} ; \mathcal{G} _ 2, \mathcal{G} _ {s1}\right)$**. Both **$\mathcal{L} _ {con1}$** and **$\mathcal{L} _ {con2}$** are alternative representations of **$I\left(\mathcal{G} _ {s1} ; \mathcal{G} _ {s2}\right)$** and **$I\left(\mathcal{G} _ {s2} ; \mathcal{G} _ {s1}\right)$** respectively, as discussed in references [3, 4, 5, 6].
> > Therefore, **Equation 16** can be seen as an approximate upper bound for **Equation 8**. To minimize **Equation 8**, we instead minimize its upper bound. Minimizing the upper bound indirectly minimizes the target loss function, as optimizing the upper bound ensures that the optimal solution for the target loss function is not overestimated, thereby effectively approaching the minimum value.

---

> > > ### Author Response · Authors · 2024-11-20
> > >
> > > > **W8.**   In Figure 4, the focus of the network substantially changes over iteration. Is that expected or is it a sign of instability?
> > >
> > > We apologize for any confusion caused by Figure 4. The reason why the focus of the network substantially changes over iterations is that the figures shown represent taken at relatively large intervals between iterations. Due to page limitations, we could only include a few key iterations, but a more detailed representation of the iterative process is shown in the appendix.
> > >
> > > As shown in **Figure 7**, we provide a more granular view of the network's focus during both the early and late stages of the iterations. In the early stages, the network fluctuates between several candidate focal points, but by the later stages, the network's focus tends to converge and stabilize. This illustrates the stability of our model after the initial fluctuations, demonstrating its reliability as the training progresses.
> > >
> > > [ 1 ] Chris J Maddison, Andriy Mnih, and Yee Whye Teh. The concrete distribution: A continuous relaxation of discrete random variables.
> > >
> > > [ 2 ] Eric Jang, Shi xiang Gu, and Ben Poole. Categorical reparameterization with gumbel-softmax.
> > >
> > > [ 3 ] Tian, Y., Krishnan, D., and Isola, P. Contrastive multiview coding.
> > >
> > > [ 4 ] Hjelm, R. D., Fedorov, A., Lavoie-Marchildon, S., Grewal, K., Bachman, P., Trischler, A., and Bengio, Y. Learning deep representations by mutual information estimation and maximization.
> > >
> > > [ 5 ] You, Y., Chen, T., Sui, Y., Chen, T., Wang, Z., and Shen, Y. Graph contrastive learning with augmentations.
> > >
> > > [ 6 ] Velickovi ˇ c, P., Fedus, W., Hamilton, W. L., Li ´ o, P., Bengio, Y., and Hjelm, R. D. Deep graph infomax.

---

> > > > ### Comment · Reviewer_WivT · 2024-11-23
> > > >
> > > > Thanks for the detailed answer to my clarifications. Please incorporate the suggestions in the final manuscript.
> > > > Overall, I believe this is a solid contribution and I am happy to increase my score.

---

> > > > > ### Author Response · Authors · 2024-11-23
> > > > > **Grateful Acknowledgment and Future Commitments**
> > > > >
> > > > > **Dear Reviewer WivT,**
> > > > >
> > > > > We sincerely appreciate your encouraging feedback and are truly heartened by your acknowledgment of our work. Your thoughtful suggestions and insights have been invaluable, and we are committed to carefully incorporating them into the final manuscript to further improve its quality.  We once again extend our heartfelt thanks for your valuable time, thoughtful effort, and insightful contributions.
> > > > >
> > > > > Warm regards,
> > > > >
> > > > > Authors

---

### Official Review · Reviewer_xkpA · 2024-11-03

**Soundness:** 3
**Presentation:** 4
**Contribution:** 3
**Rating:** 6
**Confidence:** 4

**Summary:**

To alleviate the problems in current methods of molecular relational learning: insufficient consideration of molecular interactions and failure to capture high-quality substructures, this paper introduces an IGIB (Interactive Graph Information Bottleneck)-ISE (Iterative Substructure Extraction) method. Their work achieves better performance than current SOTA models in terms of accuracy, generalizability, and interpretability.

**Strengths:**

1.	This paper has good clarity. It is well-written with a clear structure. In a concise but informative style, readers would find it easy to understand the key concepts, backgrounds, and methods.
2.	Their work also brings new insights into the MRL area. They noticed the inefficiency of current methods, where using the complete profile of an interacting molecule could not only be unnecessary but also comprises generalizability. And they proved the effectiveness of their method through experiments.
3.	In general, they bring new ideas to the MRL area: Interactive Graph Information Bottleneck (IGIB). Bottleneck-based methods are widely used in many areas and receive satisfactory results. In this paper, they integrated it into the ISE framework for further optimization. It is also the method that leverages the model’s performance to outperform all baselines.

**Weaknesses:**

1.	(General Assumption) Most molecule interactions may depend on each molecule’s substructures, but does this apply to all molecule interactions? If not, the assumption at line 161 is somewhat arbitrary, where some edge cases could be ignored by this model. This assumption needs to be further justified.
2.	(Time and Space Complexity) While the model outperforms all the baseline models, it spends much more time processing DDI Datasets. Compared to CMRL, with around 1% accuracy improvement, this model costs 5.8 ~ 7.1x more time and 6.4 ~ 9x more space. This may lead to expensive computation. The trade off between the performance and computing cost needs to be examined.
3.	 (Ablation Experiment) Most experiments are designed well, but the experiment in line 1224 is less persuasive. Among all the datasets for the drug-drug interaction prediction task, ChChMiner has the fewest data points. Besides, since molecular interaction prediction tasks are different from DDI, a separate experiment would be good.
4.	 (Improvement) While IGIB-ISE achieves good performance, ISE fails to outperform all Category II methods in Table 1 (line 324) and some Category II methods in Table II (line 378). Also, the improvement of IGIB-ISE is not that noticeable in the classification task.

**Questions:**

1. Please justify your assumption stated at line 161.
2. For Line 1224 Figure 5, why do you only choose to conduct the ablation study on the ChChMiner dataset? Ablation studies on larger datasets are needed.
3. Following your design, IGIB-ISE should effectively identify the core substructure of molecules, why did the model not improve the classification accuracy more? As it reduces redundant information, why does it occupy a larger space? More analysis is needed to identify factors that may limit the improvement. What are the potential enhancement may be introduced to address these limitations?

---

> ### Author Response · Authors · 2024-11-20
> **Response to Reviewer xkpA:**
>
> Thank you very much for your valuable comments!
> We are immensely gratified and encouraged to learn that our proposed method, the problem tackled, and our experiments have garnered your acknowledgment.
> Below, we have carefully considered and responded to your valuable comments point by point.
>
> > **W1 & Q1.** *This assumption, "Molecule interactions depend on each molecule’s substructures," needs to be further justified.*
>
> Thank you for your professional feedback.  Indeed, not all molecular interactions are solely dependent on substructures and recognize that certain types of interactions, such as van der Waals forces [ 1 ], may not directly rely on specific substructures.
> To address your concern and enhance the rigor of our discussion, we will revise line 161 in the updated manuscript to state:  *"Secondly, because most interactions between molecules arise from the interactions between their core substructures,"*
>
>
>
> > **W2.** *It spends much more time processing DDI datasets. The trade-off between performance and computing cost needs to be examined.*
>
> Thank you for your constructive feedback.
> As noted in Tables 6 and 7 of our original manuscript, our method indeed requires significantly more time to process the DDI datasets. This is primarily because of the higher number of iterations (**IN**) set for these datasets. We intentionally set higher **IN** values for these datasets to maximize accuracy, achieving state-of-the-art performance. However, we recognize the need to balance performance and computational cost. As demonstrated in **Figure 3(b)** of our paper, the performance of our model improves rapidly when **IN < 5**. Beyond **IN = 10**, the rate of performance improvement diminishes significantly. This indicates that selecting **IN = 5-10** strikes an optimal balance, achieving over **50% of the best performance** while substantially reducing computational expense.
>
> To further address your concern, we conducted additional experiments with fixed **IN = 10** across datasets, and the results are summarized below (Bold indicates the best result, italic indicates the second best result) :
>
> | Model                     | Metric | ZhangDDI   | ChChMiner  | DeepDDI    |
> | ------------------------- | ------ | ---------- | ---------- | ---------- |
> | **CGIB**                  | Memory | 5.1G       | 3.9G       | 7.4G       |
> |                           | Time   | 1.5h       | 0.6h       | 3.7h       |
> |                           | ACC    | 87.69%     | 94.68%     | 95.76%     |
> | **CMRL**                  | Memory | 4.0G       | 3.4G       | 6.1G       |
> |                           | Time   | 1.3h       | 0.5h       | 3.2h       |
> |                           | ACC    | 87.78%     | 94.43%     | 95.99%     |
> | **IGIB-ISE (Optimal IN)** | Memory | 36G        | 27G        | 39G        |
> |                           | Time   | 8.7h       | 2.9h       | 22.7h      |
> |                           | ACC    | **88.84%** | **95.56%** | **96.65%** |
> | **IGIB-ISE (IN = 10)**    | Memory | 11.7G      | 8.3G       | 10.4G      |
> |                           | Time   | 2.7h       | 0.94h      | 5.3h       |
> |                           | ACC    | *88.40%*   | *95.38%*   | *96.37%*   |
>
> From this table, it is evident that using **IN = 10** reduces memory and time consumption significantly while retaining approximately **50-80%** of the performance gains:
> - **ZhangDDI**: Memory and time consumption are reduced by **67.5%** and **69.0%**, respectively, while retaining **57%** of the performance improvement.
> - **ChChMiner**: Memory and time consumption are reduced by **69.3%** and **67.6%**, respectively, while retaining **79.5%** of the performance improvement.
> - **DeepDDI**: Memory and time consumption are reduced by **73.3%** and **76.6%**, respectively, while retaining **72.7%** of the performance improvement.
>
> The result highlights the flexibility of our method in achieving competitive performance while mitigating computational costs when needed. This adjustment underscores the trade-offs possible with our framework and addresses your concerns about cost-effectiveness.
> We will further explore and discuss these trade-offs in future work.
>
> [ 1 ] Karplus M, Kolker H J. Van der Waals forces in atoms and molecules.

---

> ### Author Response · Authors · 2024-11-20
>
> > **W3 & Q2.**  Most experiments are designed well, but ablation experiments on more datasets are helpful.
>
> Thank you for your constructive suggestion. In response, we conducted additional ablation experiments on two DDI datasets (ZhangDDI and DeepDDI) and three solvent-solute datasets (FreeSolv, Abraham, and CombiSolv). These experiments were designed to demonstrate the contribution of each model component across various data scales and task types. We ensured that all experiments followed the same setup (except for the ablated components) and repeated them five times to provide robust results. The results are reported as **Mean (Variance)**.
>  **Results on DDI Datasets (Evaluation Metric: ACC (%))**
>
> | Dataset   | $\beta _ 1 = 0$ | $\beta _ 2 = 0$ | w/o KL Loss  | w/o Contrastive Loss | Baseline         |
> | --------- | ------------- | ------------- | ------------ | -------------------- | ---------------- |
> | ZhangDDI  | 88.34 (0.41)  | 88.39 (0.27)  | 88.37 (0.39) | 88.59 (0.24)         | **88.84 (0.32)** |
> | DeepDDI   | 96.27 (0.34)  | 96.33 (0.31)  | 96.12 (0.28) | 96.41 (0.19)         | **96.65 (0.37)** |
> | ChChMiner | 94.86 (0.37)  | 94.82 (0.11)  | 94.93 (0.17) | 95.33 (0.26)         | **95.56 (0.28)** |
> **Results on Solvent-Solute Datasets (Evaluation Metric: RMSE)**
>
> | Dataset   | $\beta _ 1 = 0$ | $\beta _ 2 = 0$ | Without KL Loss | Without Contrastive Loss | Baseline          |
> | --------- | ------------- | ------------- | --------------- | ------------------------ | ----------------- |
> | FreeSolv  | 0.921 (0.058) | 0.886 (0.029) | 0.986 (0.030)   | 0.921 (0.033)            | **0.713 (0.034)** |
> | Abraham   | 0.353 (0.002) | 0.419 (0.009) | 0.414 (0.001)   | 0.366 (0.001)            | **0.343 (0.009)** |
> | CombiSolv | 0.411 (0.004) | 0.397 (0.004) | 0.413 (0.001)   | 0.411 (0.001)            | **0.394 (0.008)** |
>
> As shown in the tables, with all components active, our model achieved the best performance across all datasets. When the KL divergence loss ($\mathcal{L}{com1}$ and $\mathcal{L}{com2}$), which facilitates the compression of interactive substructures, was removed, the performance declined on all datasets, with FreeSolv and Abraham experiencing the most significant drops. This highlights the critical role of KL divergence loss in guiding the model towards more precise substructure selection, particularly in regression tasks.
>
> On the other hand, removing the contrastive loss ($\mathcal{L}{con1}$ and $\mathcal{L}{con2}$) resulted in a marginal performance reduction for most datasets, except for FreeSolv. This phenomenon could be attributed to the robust interaction modeling of our iterative interaction module, which reduces the reliance on contrastive loss. However, for the FreeSolv dataset, where fewer iterations (IN) were used, the contrastive loss played a more pivotal role, demonstrating the dataset-dependent utility of this component.
>
> Finally, we evaluated the impact of setting $\beta_1$ and $\beta_2$ to zero. For DDI datasets, the results indicate that $\beta_1$ and $\beta_2$ have similar contributions, as evidenced by the small margin of performance differences. Nevertheless, for solvent-solute datasets, $\beta_1$ and $\beta_2$ exhibited distinct impacts. This divergence may stem from the inherent asymmetry in solvent-solute interactions, suggesting that the choice of $\beta_1$ and $\beta_2$ requires careful consideration when dealing with asymmetric molecular interactions.
>
> **Ablation Design Detail**
> We based the ablation design on Appendix D.S2 of the paper. The original objective function is:
> $\underset{\mathcal{G} _ {s1}, \mathcal{G} _ {s2}}{\arg \min} -I\left(\mathbf{Y}; \mathcal{G} _ {s1}, \mathcal{G} _ {s2}\right) + \beta_1 I\left(\mathcal{G} _ 1; \mathcal{G} _ {s1} \mid \mathcal{G} _ {s2}\right) + \beta_2 I\left(\mathcal{G} _ 2; \mathcal{G} _ {s2} \mid \mathcal{G} _ {s1}\right)$ .
> - **When $\beta_1 = 0$**: The objective becomes $\underset{\mathcal{G} _ {s1}, \mathcal{G} _ {s2}}{\arg \min} -I\left(\mathbf{Y}; \mathcal{G} _ {s1}, \mathcal{G} _ {s2}\right) + \beta_2 I\left(\mathcal{G} _ 2; \mathcal{G} _ {s2} \mid \mathcal{G} _ {s1}\right)$
>   meaning $\mathcal{G} _ {s1}$ is no longer encouraged to prune or relate to $\mathcal{G} _ {s2}$.
>
> - **When $\beta_2 = 0$**: The objective becomes  $\underset{\mathcal{G} _ {s1}, \mathcal{G} _ {s2}}{\arg \min} -I\left(\mathbf{Y}; \mathcal{G} _ {s1}, \mathcal{G} _ {s2}\right) + \beta_1 I\left(\mathcal{G} _ 1; \mathcal{G} _ {s1} \mid \mathcal{G} _ {s2}\right)$
>   meaning $\mathcal{G} _ {s2}$ is no longer encouraged to prune or relate to $\mathcal{G} _ {s1}$.
>
> - **Without KL loss**: The objective becomes  $\underset{\mathcal{G} _ {s1}, \mathcal{G} _ {s2}}{\arg \min} -I\left(\mathbf{Y}; \mathcal{G} _ {s1}, \mathcal{G} _ {s2}\right) - \beta _ 1 I\left( \mathcal{G} _ {s1} ; \mathcal{G} _ {s2}\right) - \beta_2 I\left(\mathcal{G} _ {s2} ; \mathcal{G} _ {s1}\right)$

---

> ### Author Response · Authors · 2024-11-20
>
> meaning $\mathcal{G} _ {s1}$ and $\mathcal{G} _ {s2}$ are no longer encouraged to prune.
>
> - **Without Contrastive loss**: The objective becomes  $\underset{\mathcal{G} _ {s1}, \mathcal{G} _ {s2}}{\arg \min} -I\left(\mathbf{Y}; \mathcal{G} _ {s1}, \mathcal{G} _ {s2}\right) + \beta_1 I\left( \mathcal{G} _ {s1} ; \mathcal{G} _ {1}, \mathcal{G} _ {s2}\right) + \beta_2 I\left(\mathcal{G} _ {s2} ; \mathcal{G} _ {2}, \mathcal{G} _ {s1}\right)$
>   meaning $\mathcal{G} _ {s1}$ and $\mathcal{G} _ {s2}$ are no longer encouraged to be interrelated.
>
> > **W4.**  Although IGIB-ISE achieves good performance, ISE does not achieve excellent performance on some datasets.
>
> **Thank you for your insightful feedback.**
>
> In the regression task, as shown in **Table 1**, the ISE module outperforms nearly all baseline models across various tasks;  this result demonstrates the ISE module's capability to accurately extract core substructures, thereby improving performance in regression tasks. However, as for the classification task recorded in **Table 2**, the ISE module shows mixed results in **inductive settings**, with some performances not surpassing the baseline. This is likely because the ISE module does not inherently promote subgraph compactness. CGIB achieves better performance in these cases due to its integration of theoretically guided substructure pruning.  That said, the integration of IGIB theory effectively addresses this limitation, showcasing the complementary nature and applicability of IGIB and ISE. In the future, we aim to further iterate on the ISE module by incorporating substructure-scale-restrictive networks. This enhancement would reduce reliance on IGIB theory.
>
>
>
>
> > **W4 & Q3.**  The improvement of IGIB-ISE is not that noticeable in the classification tasks.
>
> Our method is designed to extract more precise interaction substructures, which enables more accurate modeling of molecular interactions. However, for classification tasks, the output is discrete class labels which is simpler than regression tasks. In many cases, even if the substructures extracted by the baseline model contain some noise, the final classification can still be effectively distinguished by certain features, resulting in good classification accuracy. Therefore, classification tasks exhibit a certain tolerance for noise, which limits the noticeable improvements of our model in some of the classification metrics.
>
> In contrast, regression tasks aim to predict continuous values, making them more sensitive to the core substructures extracted by the model. In regression tasks, even small amounts of noise can lead to significant fluctuations in predicted values, which negatively impacts the overall performance. Consequently, because our model extracts more precise interaction substructures, it shows more noticeable improvements in regression tasks.
>
>
> > **Q3.** *As it reduces redundant information, why does it occupy a larger space? More analysis is needed to identify factors that may limit the improvement. What are the potential enhancement may be introduced to address these limitations?*
>
> Thank you for your insightful question.  First, while our method effectively reduces redundant information, the increased space usage occurs only during the **training phase**. As shown in Tables 6 and 7 of the paper, our method incurs higher training time and memory overhead than baseline models due to the iterative substructure selection process integrated with the prediction module for end-to-end optimization. This results in all parameters and intermediate states produced during substructure iterations being stored in the computation graph, leading to significant overhead. Specifically:
> - **Time Complexity:** Multiple gradient computations in the interaction network account for most of the training time.
> - **Space Complexity:** Redundant storage of interaction network parameters during each iteration is the primary contributor to increased memory usage.

---

> > ### Author Response · Authors · 2024-11-20
> >
> > | Model        | Metric | ZhangDDI | ChChMiner | DeepDDI  | MNSol      | FreeSolv   | CompSol    | Abraham    | CombiSolv  |
> > | ------------ | ------ | -------- | --------- | -------- | ---------- | ---------- | ---------- | ---------- | ---------- |
> > | Data Volume  |        | 113,972  | 33,669    | 316,595  | 3,037      | 560        | 3,548      | 6,091      | 8,780      |
> > | **CGIB**     | Memory | 5.1G     | 3.9G      | 7.4G     | 2.1G       | 2.1G       | 2.1G       | 2.4G       | 2.3G       |
> > |              | TIME   | 1.5h     | 0.6h      | 3.7h     | 2.3min     | 0.2min     | 4.8min     | 9.5min     | 8.8min     |
> > | **CMRL**     | Memory | **4.0G** | **3.4G**  | **6.1G** | 2.1G       | 2.1G       | **2.1G**   | **2.4G**   | **2.3G**   |
> > |              | TIME   | **1.3h** | **0.5h**  | **3.2h** | **2.2min** | **0.2min** | **4.2min** | **8.7min** | **7.4min** |
> > | **IGIB-ISE** | Memory | 36G      | 27G       | 39G      | **2.1G**   | **1.8G**   | 2.2G       | 2.6G       | 2.4G       |
> > |              | TIME   | 8.7h     | 2.9h      | 22.7h    | 5.1min     | 0.2min     | 8.8min     | 13.0min    | 14.75min   |
> >
> >
> >
> > **Inference Phase: Time and Space Complexity**
> > In real-world applications, inference efficiency is critical. Once trained, the core substructure extractor is directly applied for molecular interaction predictions. The table below compares our model with others across various datasets (with ~1/5 sampling from the DDI dataset and all samples for Solvent-Solute Datasets):
> >
> > | Model        | Metric | ZhangDDI   | ChChMiner | DeepDDI    | MNSol     | FreeSolv  | CompSol   | Abraham   | CombiSolv |     |
> > | ------------ | ------ | ---------- | --------- | ---------- | --------- | --------- | --------- | --------- | --------- | --- |
> > | Data Volume  |        | 20,000     | 4,932     | 70,000     | 3,037     | 560       | 3,548     | 6,091     | 8,780     |     |
> > | **CGIB**     | Memory | 278M       | 303M      | 297M       | 85M       | 38M       | 85M       | 103M      | 104M      |     |
> > |              | TIME   | 24.76s     | 6.81s     | 94.67s     | 1.69s     | 0.94s     | 2.62s     | 3.64s     | 4.76s     |     |
> > | **CMRL**     | Memory | **236M**   | **254M**  | **252M**   | **72M**   | **34M**   | **71M**   | **94M**   | **92M**   |     |
> > |              | TIME   | 23.46s     | **5.89s** | 77.26s     | **1.49s** | **0.88s** | 2.31s     | **3.43s** | **4.37s** |     |
> > | **IGIB-ISE** | Memory | 275M       | 301M      | 294M       | 81M       | 37M       | 75M       | 101M      | 98M       |     |
> > |              | TIME   | **22.58s** | 5.97s     | **74.98s** | 1.62s     | 0.92s     | **2.22s** | 3.53s     | 4.55s     |     |
> >
> >
> > The experimental results demonstrate that IGIB-ISE achieves superior overall performance in terms of spatiotemporal complexity during inference. While its memory usage is comparable to that of baseline models (CGIB and CMRL), IGIB-ISE occasionally shows advantages in runtime. For example, on the ZhangDDI and DeepDDI datasets, IGIB-ISE completes inference in just 22.58s and 74.98s, respectively, outperforming both CGIB and CMRL. On smaller datasets such as ChchMiner and FreeSolv, IGIB-ISE exhibits marginally better time efficiency than CGIB and is nearly on par with CMRL, indicating excellent scalability and adaptability. Overall, IGIB-ISE does not incur significant resource overhead during inference, proving highly practical and suitable for real-world applications.
> >
> > Finally, we consider three potential engineering optimizations to improve the efficiency of **Training Phase.**
> > - **Optimization of Computation Graph Storage:** Interaction network parameters, unchanged during iterations, can be globally stored and reused for gradient computation. This reduces redundant storage while maintaining functionality.
> > - **Core Substructure Initialization:** Reducing iterations by initializing substructures based on prior chemical knowledge (e.g., functional groups) can accelerate convergence and reduce training overhead.
> > - **Efficient Parameter Fine-Tuning (e.g., LoRA [2]):** Using low-rank matrices for fine-tuning allows freezing the interaction network and adapting it with minimal computational cost, significantly reducing both memory and time requirements. The pre-trained parameters of the interaction network can be obtained from baseline models such as CGIB.
> >
> > [ 2 ] Edward J. Hu, Yanan Wu, Xuezhi Chen, et al. LoRA: Low-Rank Adaptation of Large Language Models.

---

> ### Author Response · Authors · 2024-11-23
> **Dear Reviewer xkpA,**
>
> Dear Reviewer xkpA,
>
> We noticed a potentially confusing operation. We sincerely value your perspective and would appreciate the opportunity to engage in further discussions to better understand your concerns. If you have any additional feedback or questions, we would be most grateful if you could kindly share them with us.
>
> Warm regards,
>
> The Authors

---

> ### Author Response · Authors · 2024-11-29
> **Dear Reviewer xkpA**
>
> **Dear Reviewer xkpA**,
>
> We greatly appreciate the time and effort you have devoted to reviewing our manuscript. We have carefully provided detailed responses to your comments and would like to kindly inquire whether our revisions and explanations have sufficiently addressed your concerns.
>
> If there are any remaining questions or further feedback, we would be more than happy to engage in further discussion to ensure we meet your expectations.
>
> Once again, thank you for your invaluable feedback, which has been instrumental in enhancing the quality of our work.
>
> Best regards,
> Authors

---

> > ### Comment · Reviewer_xkpA · 2024-12-02
> >
> > Thank you for providing the detailed explanation and additional experiments. I have no further question.

---

### Author Response · Authors · 2024-11-20
**# General Response to All the Reviewers**

We sincerely thank all the reviewers for their insightful and constructive feedback on our manuscript. We are delighted to hear that **our idea was recognized as novel** (all reviewers), and that **our work addresses significant issues in the field or opens new insights** ($\frac{3}{4}$ reviewers: xkpA, WivT, and spWZ). Additionally, they think that  **our experimental results are satisfactory or promising** (all reviewers).

We have carefully examined all the suggestions and provided detailed responses to each point. If there are any further questions or concerns, please do not hesitate to let us know. We are fully committed to engaging in further discussions and to improving the quality of this work. Once again, thank you for your invaluable comments!

---

### Meta-Review · Area_Chair_bLYi · 2024-12-20

**Metareview:**

This paper proposes an Interactive Graph Information Bottleneck with the Iterative Substructure Extraction method to improve molecular relational learning by focusing on molecular interactions through core substructures. The idea is considered well-motivated, with a sound solution exploiting substructure relationships and innovative use of EM algorithms. Although there are some weaknesses, such as technical clarity issues and the limited discussion of method robustness and scalability, fortunately, the authors have addressed the main issues by providing solid experimental validation.

**Additional Comments On Reviewer Discussion:**

The idea is considered well-motivated with a sound solution exploiting substructure relationships (WivT), innovative in using EM algorithms for substructure extraction as latent variables (spWZ), and bringing new insights to the MRL area through bottleneck-based methods (xkpA). Although there are some weaknesses were identified by the reviewers, such as the technical clarity issues (spWZ, WivT), and the limited discussion of method robustness and scalability (spWZ, Vr3G), fortunately, the authors have addressed the main issues by providing solid experimental validation.

---

### Decision · Program_Chairs · 2025-01-22

Accept (Poster)